# Efficient multi-scale Gaussian process regression for massive remote sensing data with satGP v0.1.2

Jouni Susiluoto[1,2,3,4], Alessio Spantini[1], Heikki Haario[2,3], Teemu Härkönen[2], and Youssef Marzouk[1]

[1]Massachusetts Institute of Technology, Department of Aeronautics and Astronautics, 77 Massachusetts Avenue, 33-207, Cambridge MA 02139 USA
[2]Lappeenranta University of Technology, School of Engineering Science, P.O. Box 20, FI-53851 Lappeenranta, Finland
[3]Finnish Meteorological Institute, Erik Palménin aukio 1, FI-00560 Helsinki, Finland
[4]Jet Propulsion Laboratory, California Institute of Technology, 4800 Oak Grove Drive, Pasadena CA 91109 USA

**Correspondence:** Jouni Susiluoto (jsusiluo@mit.edu)

**Abstract.** Satellite remote sensing provides a global view to processes on Earth that has unique benefits compared to making measurements on the ground, such as global coverage and enormous data volume. The typical downsides are spatial and temporal gaps and potentially low data quality. Meaningful statistical inference from such data requires overcoming these problems and developing efficient and robust computational tools. We design and implement a computationally efficient multi-scale Gaussian process (GP) software package, satGP, geared towards remote sensing applications. The software is able to handle problems of enormous sizes and to compute marginals and sample from the random field conditioning on at least hundreds of millions of observations. This is achieved by optimizing the computation by e.g. randomization and splitting the problem into parallel local subproblems which aggressively discard uninformative data.

We describe the mean function of the Gaussian process by approximating marginals of a Markov random field (MRF). Variability around the mean is modeled with a multi-scale covariance kernel, which consist of Matérn, exponential, and periodic components. We also demonstrate how winds can be used to inform covariances locally. The covariance kernel parameters are learned by calculating an approximate marginal maximum likelihood estimate, and the validity of both the multi-scale approach and the method used to learn the kernel parameters is verified in synthetic experiments.

We apply these techniques to a moderate size ozone data set produced by an atmospheric chemistry model, and to the very large number of observations retrieved from the Orbiting Carbon Observatory 2 (OCO-2) satellite. The satGP software is released under an open source license.

## 1 Introduction

Climate change is one of the most important present-day global environmental challenges. The underlying reason is anthropogenic carbon emissions. According to the Intergovernmental Panel on Climate Change, carbon dioxide (CO2) has the

strongest effect on warming the planet of the well-mixed greenhouse gases, with the radiative forcing of ca. 1.68 W m$^{-2}$ (IPCC, 2013).

Several instruments orbiting the Earth produce enormous quantities of remote sensing data, used to compute local estimates of CO2 and other atmospheric constituents by solving complicated inverse problems, and further processed to e.g. gridded data products and flux estimates (Cressie, 2018). These instruments include the Greenhouse gases Observing SATellite (GOSAT) from Japan (Yokota et al., 2009), operational since January 2009, the OCO-2 from NASA (Crisp et al., 2012), launched in July 2014, and the Chinese TanSat (Yi et al., 2018), launched in December 2016. GOSAT-2 was launched in October 2018, and in May 2019 the OCO-3 instrument (Eldering et al., 2019) was taken to the International Space Station. In addition to the CO2-measuring instruments, also other types of data are produced by remote sensing. For instance the European TROPOspheric Monitoring Instrument (TROPOMI) produces measurements of nitrogen dioxide, formaldehyde, carbon monoxide, aerosols, methane, and ozone. Common denominators among most non-gridded remote sensing data sets include a large number of observations, global coverage but small area observed at any given time, sensitivity to prevailing weather conditions and cloud cover, unknown and/or unreported error covariances, and predetermined positioning that rules out freely observing at a given time and location. These shortcomings can be partly remedied with techniques from computational statistics, such as those implemented in the satGP software, which this paper introduces.

There are two key advances in this work. First, we describe the computational approaches that allow satGP to tackle remote-sensing related spatial statistics problems of enormous sizes. Second, we present formulations of a multi-scale covariance function and a space-dependent mean function, types of which we have not seen used in the remote sensing community. We also show how these functions can be efficiently learned from data.

Related to this work, several kriging studies have been published before in the context of remotely sensed CO2. Zeng et al. (2013) analyzed the variability of CO2 in both space and time over China and produced monthly maps from GOSAT data with slightly over 10000 observations. Nguyen et al. (2014) used a four times larger set of observations with Kalman Smoothing in a reduced dimension with GOSAT and the Atmospheric InfraRed Sounder (AIRS) data from NASA. A map of atmospheric carbon dioxide derived from GOSAT data was presented at the higher resolution of $1 \times 1.25$ degrees in space and 6 days in time by Hammerling et al. (2012). In another publication by the same authors, synthetic OCO-2 observations were considered with the same spatial resolution.

More recently Zeng et al. (2017) presented a global dataset derived from GOSAT with the spatio-temporal resolution of three days and one degree, and this study evaluated also the temporal trend of the XCO2. The results were validated against observations from the Total Carbon Column Observing Network (TCCON) and against modeling results from CarbonTracker and Goddard Earth Observing System with atmospheric chemistry (GEOS-Chem). Tadić et al. (2017) described a moving window block kriging algorithm to introduce time dependence into a GOSAT-based XCO2 map construction process using a quasi-probabilistic screening method for subsampling observations, thinning the data for computational reasons. Other recent studies have also contained analyses of OCO-2 data – for example Zammit-Mangion et al. (2018) presented fixed rank kriging results based on OCO-2 data using a 16-day moving window. In many of these studies, the obtained CO2 fields appear very smooth.

Applications to remote sensing data have also resulted in publications more focused on methods. Ma and Kang (2017) described a "fused" Gaussian process, combining a graphical model with a Gaussian process and applying that to sea surface temperature data. In another computationally sophisticated application, Zammit-Mangion et al. (2015) simultaneously modeled both flux fields and concentrations using a bivariate spatio-temporal model with Hamiltonian Monte Carlo (Neal, 2011) for sampling the posterior. Due to computational challenges the spatial area investigated in this work was very small.

For Gaussian processes, various approaches have been studied to overcome the difficulties posed by large amounts of data. For instance, Lindgren et al. (2011) provide an explicit link between some random fields arising as solutions to certain stochastic partial differential equations and Markov random fields. A recent review of Vecchia-type approximations (Vecchia, 1988) is given by Katzfuss et al. (2018), and Heaton et al. (2018) presents a comparison of the performance of several recently developed spatial statistics methods with applications to data from the Moderate-resolution imaging spectroradiometer (MODIS). The difficulty of ordering the observations for effective inference with Gaussian processes, especially as the dimension of the inputs grows, is discussed by Ambikasaran et al. (2016).

In this work we describe the satGP program that solves very large spatio-temporal statistics problems with up to at least the order of $10^8$ marginals conditioned on $10^8$ observations. While advances have recently been made in the field, we are not aware of any literature or software solving problems of quite this scale so far. The effectiveness is partly based on combining ideas related to Vecchia-type and nearest neighbor Gaussian processes (Datta et al., 2016), but satGP also employs several computational tricks such as subsampling observations and filtering out uninformative data at several levels when possible. The program includes a flexible implementation for space-dependent mean functions and space-independent covariance kernels, and routines for learning their parameters from data. The spatial dependence of the mean function is learned by computing marginals of a Markov random field (MRF). The covariance function is constructed in a way that allows for describing the multiple natural length scales in the data. After learning the model parameters the program computes posterior predictive fields, and realizations can be drawn from both the posterior and the prior.

We validate the multi-scale covariance modeling approach by learning the covariance function parameters of a data set drawn with satGP from the prior of a multi-scale Gaussian process. To demonstrate the computational capabilities of this early version satGP, we computed global XCO2 concentrations for a duration of 1526 days at $0.5°$ spatial and daily temporal resolution, amounting to calculating 350 million marginal distributions, conditioning on 116 million XCO2 observations from OCO-2. Figure 9 shows an example of how these results look like. We also present a non-stationary covariance kernel formulation that utilizes wind data for computation, and use that covariance function with OCO-2 data. The utility of using winds with CO2 data has been demonstrated before by e.g. Nassar et al. (2017).

In addition to the OCO-2 work we demonstrate the capabilities of satGP with synthetic ozone data from the Whole Atmosphere Community Climate Model (WACCM4) (Marsh et al., 2013), emulating observing with the Global Ozone Monitoring by Occultation of Stars (GOMOS) instrument (Bertaux et al., 2004, 2010; Kyrölä et al., 2004) on the Envisat satellite. Using synthetic data allows us to directly compare Gaussian process posterior estimates to an exactly known ground truth. The software could equally well be applied to any other observed quantity of interest.

The rest of the manuscript is organized in the following manner: Section 2 describes the methods both generally and as implemented in satGP. Section 3 discusses the computational details in satGP. Section 4 presents and discusses simulation results, including a multi-scale synthetic parameter identifiability study, an application to synthetic WACCM4-generated data, and applications using the OCO-2 v9 data. In the concluding Sect. 5 current limitations and some possible future directions are briefly mentioned.

## 2 Methods

In geosciences, kriging (Cressie and Wikle, 2001; Chiles and Delfiner, 2012) is used for performing spatial statistics tasks such as gap-filling or representing data in a grid. The semivariogram models used in kriging are closely related to the covariance models used in the Gaussian process formalism (Santner et al., 2003; Rasmussen and Williams, 2006; Gelman et al., 2013), where instead of learning the variogram model from the data, a form of a covariance function is prescribed and its parameters estimated.

With Gaussian processes, we want to learn properties of a spatio-temporal surface from some observational data of some quantity of interest. To each point in space and time corresponds a Gaussian distribution of that quantity, whose mean and variance can be calculated by solving a local regression problem. This is closely related to solving a spatio-temporal interpolation problem when the observations have Gaussian errors.

The theory of Bayesian statistics, Gaussian processes, and Markov random fields that is used in this work is well known and therefore many of the the novel aspects in this section have to do with the computational methods and modifications that are presented, such as observation selection schemes in Sect. 2.5 or approximate marginal maximum likelihood computation in Sect. 2.6. These modifications trade precision for tractability, but in a way that tries minimize the loss in accuracy. Due to the desire to be able to solve very large problems, some sacrifices need to be made to be able to obtain any solution.

This section goes through the Gaussian process formalism and presents both generic and satGP-specific forms of mean and covariance functions. This is followed by discussion of how observation selection is carried out for solving local subproblems and how model parameters are learned. The presentation of the general Gaussian process problem is based on Santner et al. (2003) and Rasmussen and Williams (2006). Commonly used notation is listed in Table 1.

### 2.1 Gaussian process regression

A Gaussian process is a stochastic process, which can be thought of as an infinite-dimensional Gaussian distribution in that the joint distributions of the process at any finite set of space-time points are multivariate normal. We denote points in the spatio-temporal domain by $x \in \mathbb{R}^q$. In this work $q = 3$, even though this restriction can be overcome if needed, and satGP does have limited support for space-only problems.

The Gaussian process, or Gaussian random field, is denoted by

$$\Psi \sim \mathrm{GP}(m(x; \beta), k(x, x'; \theta)), \tag{1}$$

where $m : \mathbb{R}^q \to \mathbb{R}$ and $k : \mathbb{R}^{q \times q} \to \mathbb{R}$ are the mean and covariance functions of the process parameterized by hyperparameter vectors $\beta \in \mathbb{R}^{n_\beta}$ and $\theta \in \mathbb{R}^{n_\theta}$. The infinite-dimensional (since the domain of $x$ is typically infinite) description in Eq. (1) is below reduced to a finite-dimensional problem, in which case $k(x, x')$ describes an entry of the covariance matrix of the joint distribution of random variables $\Psi(x)$ over all $x$ that one is interested in.

The function $m$ above is called drift in kriging literature, and the expected value of the process in regions with no data will tend to the value of this mean function. It is chosen to reflect the deterministic patterns in the data, and the particular form picked to model $m$ will also affect how the function $k$ and parameters $\theta$ in Eq. (1) need to be specified. With inadequate modeling of the mean function, the uncertainty estimates obtained with Gaussian process regression may end up being unnecessarily large. For instance linear trends, constant factors, seasonal and other periodic fluctuations should be included in $m$ if they are known.

An example of what is used with the OCO-2 data is shown later in Eq. (11).

In what follows, the domain $\mathbb{R}^q \ni x$ is divided into two disjoint parts, one of which, $\mathcal{X}^{\mathrm{train}} \subset \mathbb{R}^q$, is the set of coordinates $x_i^{\mathrm{obs}}$, where observation data (training data) were measured, and another one, $\mathcal{X}^{\mathrm{test}} \triangleq \mathbb{R}^q \backslash \mathcal{X}^{\mathrm{train}}$, denotes its complement. Points in $\mathcal{X}^{\mathrm{test}}$ are denoted by $x^*$ and called *test inputs* as is often done in Gaussian process literature. Observations at locations $\{ x_i^{\mathrm{obs}} : i = 1, \dots, n \}$, both real and synthetic ones generated by the Gaussian process, are denoted by $\psi_i^{\mathrm{obs}} \in \mathbb{R}$, and the vector

of all $\psi_i^{\mathrm{obs}}$ is written $\psi^{\mathrm{obs}}$.

For the mean function $m$ in Eq. (1) a specific form,

$$m(x; \beta, \delta) = f(x; \delta)^T \beta \equiv \widetilde{f}(x^{\mathrm{t}}; \delta(x^{\mathrm{s}}))^T \beta(x^{\mathrm{s}}), \tag{2}$$

is used in this work. The superindexes s and t refer to the spatial and temporal parts of the generic coordinate $x$, and $\delta$ are auxiliary parameters which are space-dependent. The purpose of the right hand side with the function $\widetilde{f}$ is to underline that $f$

depends on the spatial part of $x$ only via the space-dependent $\delta$ parameters, and that the $\beta$ parameters do not depend on $x^{\mathrm{t}}$, the temporal part of $x$. The temporal evolution of the mean function is in this particular form determined only by the function $f(x; \delta) \triangleq [f_1(x; \delta), \dots, f_{n_\beta}(x; \delta)]^T$, and for each $f_i$ there is a space-dependent regression coefficient $\beta_i$.

The parameter vectors $\delta$ contains space-dependent parameters that affect the form of any of the $f_i$ in a way that cannot be modeled with the $\beta$ coefficients in the functional form of Eq. (2). The length of these space-dependent $\delta$-vectors is $n_\delta$.

Given the parameters $\delta$ for all the inputs in $x^{\mathrm{obs}}$ and a set of functions $f_i$ for constructing the mean function, we define matrix $F \in \mathbb{R}^{n \times n_\beta}$ with elements $F_{ij} = f_i(x_j^{\mathrm{obs}}; \delta)$, where the $\delta$ is now specific to the location $x_i^{\mathrm{obs}}$.

The definition of $m$ above is very general and can describe in practice a large number of realistic scenarios. Nonetheless, the form of Eq. (2) imposes the strong assumption of separation of space and time in that the $\beta$ and $\delta$ parameters do not depend on time. The explicit form of functions $f_i$ used to model the OCO-2 data are given below in Sect. 2.2.

The covariance function $k(x, x'; \theta)$ controls the smoothness of the draws $\psi$ from $\Psi$. It outputs the prior covariance of the random variables $\Psi(x)$ and $\Psi(x')$ at $x$ and $x'$. The parameter vector $\theta$ typically contains at least one scale parameter $\ell$ and a parameter $\tau$ controlling the maximum covariance, $\tau^2$. The $\ell$ parameters correspond to the length scales of the random fluctuations of the realizations around the mean function, and the $\tau$ parameters describe the amplitude of that fluctuation. By defining the covariance matrix $K \in \mathbb{R}^{n \times n}$ with elements $K_{i,j} = k(x_i^{\mathrm{obs}}, x_j^{\mathrm{obs}}; \theta)$, the joint distribution of the field at observed

locations is given by

$$\Psi^{\mathrm{obs}} \sim \mathcal{N}\left(F\beta, K\right). \tag{3}$$

Explicit forms of functions $m$ and $k$ are described in Sect. 2.2 and 2.4, respectively. Additional practical guidelines are given in Appendix A.

Bayesian statistics is a standard paradigm for analyzing data and uncertainties, and it is also widely used in geosciences (Rodgers, 2000; Gelman et al., 2013). Given the observed data $\Psi^{\mathrm{obs}} = \psi^{\mathrm{obs}}$ at some finite set of points $x^{\mathrm{obs}}$, the object of interest of the Bayesian inference problem in this work is the joint posterior distribution of the Gaussian process and the parameters,

$$p(\psi,\beta,\delta,\theta|\psi^{\mathrm{obs}}) = \frac{p(\psi^{\mathrm{obs}}|\psi,\beta,\delta,\theta)p(\psi|\beta,\delta,\theta)p(\beta,\delta,\theta)}{p(\psi^{\mathrm{obs}})}, \tag{4}$$

where $p(\psi|\beta,\delta,\theta)$ is the Gaussian process prior and $p(\beta,\delta,\theta)$ is a prior on the Gaussian process hyperparameters. In this particular equation $\beta$ and $\delta$ actually denote spatially varying hyperparameter fields. The calculation in Eq. (4) is not generally tractable for a huge number of inputs $x$, but posterior estimates of the GP, $p(\psi|\psi^{\mathrm{obs}},\hat{\beta},\hat{\delta},\hat{\theta})$, can be calculated for a finite set of inputs by conditioning on parameter point estimates $\hat{\theta}$, $\hat{\beta}$, and $\hat{\delta}$. The covariance parameter estimate $\hat{\theta}$ may be found by minimizing some loss function $\mathcal{L}$,

$$\hat{\theta} = \arg\min_{\theta} \mathcal{L}(\theta), \tag{5}$$

described explicitly below in Sect. 2.6. Given a point estimate of the parameters $\theta$ and $\delta$, and a Gaussian prior for the $\beta$ parameters with mean $\mu_\beta$ and covariance $\Sigma_\beta$, the posterior distribution of the $\beta$ parameters can be computed with,

$$\mathbb{E}[\beta|\Psi^{\mathrm{obs}} = \psi^{\mathrm{obs}}, \theta, \delta] = (F^T K^{-1} F + \Sigma_\beta^{-1})^{-1} F^T K^{-1} (\psi^{\mathrm{obs}} - F\mu_\beta) + \mu_\beta \tag{6}$$

$$\mathrm{Cov}[\beta|\Psi^{\mathrm{obs}} = \psi^{\mathrm{obs}}, \theta, \delta] = (F^T K^{-1} F + \Sigma_\beta^{-1})^{-1}. \tag{7}$$

The matrix $K$ is generally a dense matrix of size $n \times n$, where $n$ is the number of observations, and as $n$ may be extremely large, direct inversion of this matrix is in practice impossible. However, in this work inverting the full $K$ is not necessary: we want to find parameters $\beta$ that vary locally, which is done by splitting the full problem into many smaller subproblems, solving the $\beta$ parameters in a grid, as described in Sect. 2.3. This grid is then used to construct matrix $F$ by interpolating the values of $\beta$ and $\delta$ obtained.

The $\delta$ parameters are found approximately in this work by a three-step process: first a point estimate of parameters $\beta$ and $\delta$ is computed using an optimization algorithm, second, parameters $\beta$ are re-computed by Eq. (6) given the estimate of $\delta$ from the first stage, and third, the $\delta$ parameters alone are re-calibrated by optimization using the newly found $\beta$ parameters. In practice this procedure produces stable results with the OCO-2 data, and for pathological data sets repeated alternating optimization of the parameters may be performed. The calibration process is described in more detail in Sect. 2.3.2.

Even though a full posterior distribution of the parameters is not obtained this way, the solution of the Gaussian process itself is Bayesian in that the posterior marginals at each $x^*$ are found by conditioning on the observations. In the satGP software, the space-dependent $\beta$ and $\delta$ parameters are fitted first, and any learning of the covariance parameters is done only after that.

For prediction in the context of Gaussian random functions, the properties of multivariate normal distributions are exploited for calculating marginals of the random field $\Psi$ at any set of inputs. The posterior distribution $p(\psi^* | \psi^{\mathrm{obs}}, \hat{\theta}, \hat{\beta})$ of the Gaussian process at some test input $x^*$ can, given point estimates $\hat{\beta}$ and $\hat{\theta}$, be modeled according to Eq. (3) with

$$
\begin{pmatrix} \Psi^* \\ \Psi^{\mathrm{obs}} \end{pmatrix} \sim \mathcal{N}\left( \begin{bmatrix} f(x^*)^T \\ F \end{bmatrix} \hat{\beta}, \begin{bmatrix} K(x^*, x^*) & K(x^*, x^{\mathrm{obs}}) \\ K(x^{\mathrm{obs}}, x^*) & K(x^{\mathrm{obs}}, x^{\mathrm{obs}}) \end{bmatrix} \right),
\tag{8}
$$

where the vector of inputs has been divided into two parts – one for the test input $x^*$, and the other one for the observations $x^{\mathrm{obs}}$. The notation $K(x^*, x^{\mathrm{obs}})$ refers to the first row (minus the first element) of the covariance matrix with elements $K(x^*, x^{\mathrm{obs}})_j = k(x^*, x_j^{\mathrm{obs}})$, and the matrix in the lower right corner, $K(x^{\mathrm{obs}}, x^{\mathrm{obs}})$ is the same as matrix $K$ in e.g. Eq. (3). The random variable at $x^*$ can then be written as $\Psi^* | \hat{\beta}, \hat{\theta} \sim \mathcal{N}(\mu^*, \Sigma^*)$, where its mean and covariance are given by

$$
\mu^* = f(x^*)^T \hat{\beta} + K(x^*, x^{\mathrm{obs}}) K(x^{\mathrm{obs}}, x^{\mathrm{obs}})^{-1} (\psi^{\mathrm{obs}} - F\hat{\beta})
\tag{9}
$$

and

$$
\Sigma^* = K(x^*, x^*) - K(x^*, x^{\mathrm{obs}}) K(x^{\mathrm{obs}}, x^{\mathrm{obs}})^{-1} K(x^{\mathrm{obs}}, x^*),
\tag{10}
$$

and where the covariance $\Sigma^*$ is the Schur complement of $K(x^*, x^*)$. The formulas in Eq. (8) - Eq. 10 work equally well when the $x^*$ contains more than one test input. However, as of now, in satGP these equations are solved for single test input at a time. When computing $\Psi^*$ with these formulas, satGP uses observations close to $x^*$ (see Sect. 2.5), and the values of $\beta$ and $\delta$
calibrated at $x^{*\mathrm{s}}$.

## 2.2  Mean functions in satGP

Equation (2) gives the most general mean function form available in satGP. The functions $f_i$ above are user-defined and, for ease of use, satGP includes functionality for using a zero mean function, a spatially independent mean function, and an arbitrary gridded array of values. The specific forms of $f_i$ used for the OCO-2 experiments in Sect. 4 are

$$
\left.
\begin{aligned}
f_1(x) &= \sin\left(2\pi x^{\mathrm{t}} \Delta_{\mathrm{period}}^{-1} + \delta\right) \\
f_2(x) &= \cos\left(4\pi x^{\mathrm{t}} \Delta_{\mathrm{period}}^{-1} + \delta\right) \\
f_3(x) &= 1 \\
f_4(x) &= x^{\mathrm{t}}
\end{aligned}
\right\}
\tag{11}
$$

where $\Delta_{\mathrm{period}}$ is for OCO-2 the duration of one year, and $\delta$ is a space-dependent phase shift. The function $f_1$ fits the summer-winter cycle, and $f_2$ fits the semiannual cycle. It is assumed that for any given $x$, $f_1$ and $f_2$ can be modeled with the same $\delta$ parameters. The constant term is given by $f_3$, and $f_4$ gives the slow global trend. As an example of the local behavior, Fig. 1 shows the mean function fit to the observed local daily mean values of XCO2 from OCO-2 for several locations. The WACCM4
ozone study in Sect. 4.2 added two more functions $f_5$ and $f_6$ similar to $f_1$ and $f_2$, but with different $\Delta_{\mathrm{period}}$ parameters.

**Table 1.** Most commonly used notation related to inputs and mean/covariance functions in Sect. 2 and the Markov random field in Sect. 2.3.1. The second column gives the set in which the symbol belongs, or in some case the set that the symbol is a subset of. The domain sets in the second column are defined as $D_{\text{lat}} \triangleq [\text{lat}_{\min}, \text{lat}_{\max}]$, $D_{\text{lon}} \triangleq [\text{lon}_{\min}, \text{lon}_{\max}]$, $D_{\text{t}} \triangleq \mathbb{R}^{+}$, and $D \triangleq D_{\text{lat}} \times D_{\text{lon}} \times D_{\text{t}} \subset \mathbb{R}^{q}$, and $\mathcal{V}$ denotes the set of nodes in the graph described in Sect. 2.3.1.

| Symbol | $\in$ | Meaning |
|---|---|---|
| $x$ | $D$ | Generic spatio-temporal coordinate vector |
| $x^{\text{t}}$ | $D_{\text{t}}$ | Temporal part of coordinate vector $x$, implemented as seconds since 1970 |
| $x^{\text{s}}$ | $\mathbb{R}^{q-1}$ | Spatial part of generic coordinate $x$, in practice $x^{\text{s}} = [x^{\text{lat}}, x^{\text{lon}}]^{T}$ |
| $x^{\text{lat}}$ | $D_{\text{lat}}$ | North-south component of coordinate vector $x$ as defined by variable area in Table 2 |
| $x^{\text{lon}}$ | $D_{\text{lon}}$ | East-west component of coordinate vector $x$ as defined by variable area in Table 2 |
| $x^{ij}$ | $D_{\text{lat}} \times D_{\text{lon}}$ | Spatial location corresponding to $i^{\text{th}}$ latitude and $j^{\text{th}}$ longitude in the satGP regular grid |
| $x^{*}$ | $\mathbb{R}^{q}$ | Gaussian process test input – the spatio-temporal location where the GP is evaluated |
| $x^{\text{obs}}$ | $\mathbb{R}^{n \times q}$ | Matrix of space-time locations where the $n$ observations in $\psi^{\text{obs}}$ were made |
| $\beta$ | $\mathbb{R}^{n_{\beta}}$ | Mean function coefficients, see $m$ below. May be space-dependent. |
| $\beta_{\nu}$ | $\mathbb{R}^{n_{\beta}}$ | $\beta$ coefficients for the spatial location corresponding to graph label $\nu$ in the MRF |
| $\beta^{ij}$ | $\mathbb{R}^{n_{\beta}}$ | $\beta$ coefficients at grid point $x^{ij}$ in the satGP latitude-longitude grid |
| $\beta_{\mathcal{V}}$ | $\mathbb{R}^{n_{\beta} \times \mathcal{V}}$ | $\beta$ coefficients for all grid points in the satGP latitude-longitude grid |
| $\delta$ | $\mathbb{R}^{n_{\delta}}$ | Space-dependent mean function parameters that cannot be learned via Eq. (6) and (7) |
| $\delta_{\nu}$ | $\mathbb{R}^{n_{\delta}}$ | $\delta$ parameters for the spatial location corresponding to graph label $\nu$ in the MRF |
| $\delta_{\mathcal{V}}$ | $\mathbb{R}^{n_{\delta} \times \mathcal{V}}$ | $\delta$ coefficients for all grid points in the satGP latitude-longitude grid |
| $\theta$ | $\mathbb{R}^{n_{\theta}}$ | Covariance function parameters of all the subkernels of the multi-scale kernel |
| $\theta_{(\cdot)}$ | $\mathbb{R}^{n_{\theta_{(\cdot)}}}$ | Covariance function parameters of the subkernel in the subindex $(\cdot)$ |
| $I$ | - | The set of all spatial/temporal indexes for each $x$; size of $|I|$ is therefore $q$. |
| $I_{ST}$ | $\subseteq I$ | Spatio-temporal index set: corresponding $k$ is a function of space and time. |
| $I_{S}$ | $\subseteq I$ | Spatial index set: corresponding $k$ is a function of space only. |
| $\ell_{c}, c \in I$ | $\mathbb{R}^{+}$ | Covariance kernel length-scale parameter along axis $c$ |
| $\ell_{I'}$ | $\mathbb{R}^{+|I'|}$ | Covariance kernel length-scale parameters along all dimensions in $I'$ |
| $\nu$ | $\mathcal{V}$ | Label of a specific node of the graph describing the MRF. In Sect. 2.4 $\nu$ is a parameter ($\in \mathbb{R}^{+}$) used to define the Matérn kernel smoothness parameter. |
| $\nu^{ij}$ | $\mathcal{V}$ | Label of the node of the graph corresponding to the spatial location of $x^{ij}$ |
| $\partial \nu$ | $\subseteq \mathcal{V}$ | Set of nodes in the graph with edges to node $\nu$ |
| $\Psi$ | - | Random field of the quantity of interest |
| $\Psi(x)$ | $\mathbb{R}$ | Random variable of the quantity of interest corresponding to $\Psi$ at $x$ |
| $\psi^{\text{obs}}$ | $\mathbb{R}^{n}$ | Values of the observations of the field at locations $x^{\text{obs}}$ |
| $\psi$ | $\mathbb{R}^{D}$ | Realization of the random field $\Psi$ |
| $k(x, x')$ | $\mathbb{R}$ | Covariance function value of inputs $x$ and $x'$ |
| $m(x; \beta, \delta)$ | $\mathbb{R}$ | Mean function value at $x$ with parameters $\beta$ and $\delta$, $m(x; \beta, \delta) = f(x; \delta)^{T} \beta$ |
| $f(x; \delta)$ | $\mathbb{R}$ | Vector of functions to construct the mean function at $x$ with parameters $\beta$ and $\delta$ |
| $F$ | $\mathbb{R}^{n \times n_{\beta}}$ | Matrix with coefficients $F_{ij} = f_{j}(x_{i}^{\text{obs}; \delta})$ |
| $K$ | $\mathbb{R}^{n \times n}$ | Covariance matrix with elements $K_{ij} = k(x_{i}^{\text{obs}}, x_{j}^{\text{obs}})$ |

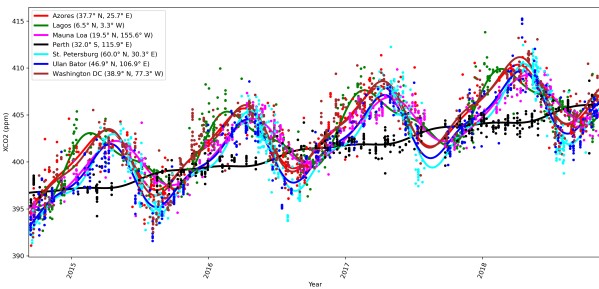

**Figure 1.** Mean function $m$ with components $f_i$ given by Eq. (11). The solid lines show the mean function value for each day, fitted to the XCO2 observations, marked by the dots. The OCO-2 mean function results are discussed in Sect. 4.4.

## 2.3 Learning the spatial dependence of $\beta$

When satGP is not used for learning GP covariance parameters or generating synthetic training sets, the finite set of test inputs $x^*$ for GP calculation is a grid with predefined geographical and temporal extents and resolution. Solving the GP marginalization and sampling problems then amounts to solving Eq. (9) and (10) at each corresponding space-time point. Since e.g. sources, sinks and timing of seasons are local, the mean function should be different from one spatial grid point to another. This is achieved by modeling the $\beta$ parameters as a Markov random field. The MRF imposes the condition that neighboring grid cells should not be too different from each other. How different they are allowed to be is a modeling choice, see Appendix A. This section describes how the spatial dependence is resolved in satGP using computational statistics.

In addition to solving this spatial problem, the marginal distributions of the $\beta$ parameters need to be solved for each individual vertex. Point estimates of the $\delta$ parameters, mentioned in Sect. 2.1, are found at the same time with the $\beta$ parameters. The intimately connected spatial and local problems are described in the subsections below.

### 2.3.1 Mean function parameters $\beta$ are described as a Markov random field

A Markov random field is a probabilistic model that describes the conditional independence structure in a set of random variables. In satGP, an MRF is used to describe how the $\beta$ coefficients depend on each other spatially. The MRF used in satGP assumes, that in addition to data, the $\beta$ coefficients only depend on the coefficient values in the neighboring grid points.

Technically, the MRF in satGP is an undirected graphical model $\mathcal{G} \triangleq (\mathcal{V}, \mathcal{E})$ (Lauritzen, 1996), with the set of vertices or nodes $\mathcal{V} \triangleq \{\nu^{ij}|i=1\ldots n_{\text{lat}}, j=1\ldots n_{\text{lon}}\}$ and edges $\mathcal{E} \triangleq \{(\nu^{i,j}, \nu^{i+1,j})|i=1\ldots n_{\text{lat}}-1, j=1\ldots n_{\text{lon}}\} \cup \{(\nu^{k,l}, \nu^{k,l+1})|k=1\ldots n_{\text{lat}}, l=1\ldots n_{\text{lon}}-1\}$. We use both $\nu$ and $\nu^{ij}$ to denote a generic vertex in a graph, and in the specific MRF setting used in satGP, each $\nu^{ij}$ corresponds to the random vector $\beta^{ij}$ at grid point $(i,j)$. After finding the marginal distributions of these vectors in the graph the *maximum a posteriori* (MAP) values of $\beta^{ij}$ are used as the parameters of the mean function for the spatial location corresponding to the $(i,j)$ element.

The set of edges $\mathcal{E}$ defines the Markov structure of the graph, i.e. how the $\beta$ coefficients of the nodes depend on each other. For any non-edge vertex $\nu^{i,j}$ there are edges in $\mathcal{E}$ to east, south, west, and north, meaning that only these neighbor-

ing vertices, collectively denoted by $\partial_{\nu^{ij}} \triangleq \{\nu \in \mathcal{V} | (\nu, \nu^{ij}) \in \mathcal{E}\}$, directly affect the vertex. More specifically, the Markov property defined by the set $\mathcal{E}$ implies that the probability of the $\beta$ parameters of latitude $i$ and longitude $j$ is given by $p(\beta^{ij}) = \int_{\partial_{\nu^{ij}}} p(\nu^{ij} | \partial_{\nu^{ij}}) p(\partial_{\nu^{ij}})$, where it is understood that $\nu^{ij}$ and $\partial_{\nu^{ij}}$ refer directly to the random variables, $\beta^{ij}$ and the joint distribution of the $\beta$ coefficients of its adjacent vertices, respectively.

The satGP program needs to compute the marginal distributions of each $\beta^{ij}$ to learn the spatially-varying mean function parameters. Due to the lattice structure of the graph, according to Hammersley and Clifford (1971) the full joint distribution of the graph $p(\mathcal{V})$ factors as $\prod_{(\nu,\nu') \in \mathcal{E}} \frac{1}{Z} \phi(\nu, \nu')$, where $Z$ is called a partition function and $\phi$ are so-called compatibility functions. This suggests that an algorithm that solves local subproblems could be used. One possible choice is the variable elimination algorithm, which is an exact standard algorithm suitable for undirected graphs of moderate size. To make the

computation faster, satGP currently modifies it by computing each diagonal in the graph, shown in Fig. 2, in parallel from $\nu^{0,0}$ to $\nu^{n_{\text{lat}},n_{\text{lon}}}$ and then back from $\nu^{n_{\text{lat}},n_{\text{lon}}}$ to $\nu^{0,0}$. Each $\nu_{ij}$ is conditioned on the previously evaluated vertices in $\partial \nu^{ij}$, but the diagonal edges of the so-called reconstituted graph are not introduced, as would normally be done. When starting again from the bottom right corner after computing diagonals numbered $1 \ldots N$, the $(N+1)^{\text{th}}$ diagonal is not conditioned on previously computed nodes. Once the diagonals $n_{\text{lon}}$ and $N + n_{\text{lon}} - 2$ that "sandwich" the node $\nu$ from both upper left and lower right

sides have been computed, the posterior distribution of $\beta_\nu$ — and any other vertex on the $(N + n_{\text{lon}} - 1)^{\text{th}}$ diagonal — can be calculated.

The modification of the algorithm loses the ability of the upper right and lower left corners to communicate effectively, but since most remote sensing data sets contain at least some observations for some time period for most nodes, the far-away information does not affect results in many practical scenarios. Techniques such as generalized belief propagation (Wainwright

and Jordan, 2008) could be used to obtain a better fit to the data, in case a need emerges to improve the spatial fitting of the mean function coefficients.

The results should not change due to changes in the user-chosen grid resolution, and for this reason satGP inversely weights the edges exponentially according to the distances between the (geographical) coordinates corresponding to the connected nodes. This rate of exponential decay is user-configurable by the `dscale` parameter, see Appendix A.

## 2.3.2   Computing the individual posterior marginals $p(\beta_\nu | \psi^{\text{obs}}, \theta)$

Assume that for the vertex $\nu$ in Fig. 2 the neighbors marked $\partial_\nu$ have been computed. Computing the marginal distribution of $\beta$ and an estimate of $\delta$ at $\nu$, referred to below as $\beta_\nu$ and $\delta_\nu$, is carried out in several steps. These steps take place inside solving the spatial problem described above: the steps listed below are computed for each vertex, corresponding to a spatial location. The computation uses information from previously computed points as prior information.

In the particular form of the mean function $m$ used for OCO-2 data in Eq. (11), the phase-shift parameter $\delta$ cannot be estimated with regression the way $\beta$ is found in Eq. (9) and (10). For this reason, the nonlinear space-dependent $\delta$-parameters are found with an optimization algorithm from the NLOpt package (Johnson, 2014), by default the BFGS algorithm, before finding $\hat{\beta}$ with Eq. (6). After obtaining $\hat{\beta}$ the $\delta$ parameters are re-optimized given the $\hat{\beta}$. The full calibration process for a single graph node $\nu$ proceeds in the following manner:

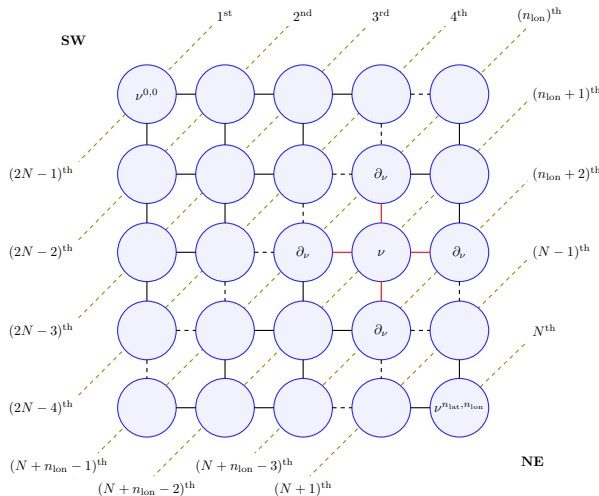

**Figure 2.** The marginal distribution of vertex $\nu$, $p(\nu)$, is conditional only on the neighbors in $\partial_\nu$ (connected to $\nu$ with red edges) due to the Markov structure in the pictured lattice graph. For effective solving, the vertices on the diagonal dashed lines are computed simultaneously making the algorithm non-exact. The order numbers labeling the diagonal lines represent an ordering in which the diagonals can be computed in parallel to get all the marginals in $\mathcal{O}(N)$ wall time, where $N \triangleq n_{\text{lat}} + n_{\text{lon}} - 1$. Southwest and northeast corners of the domain are labeled SW and NE in the graph. The final values of the parameters are obtained when diagonals from $N$ to $2N - 1$ are computed.

1. Select $n_\nu$ observations $\psi_\nu^{\text{obs}}$ of the observable that are close in terms of the spatial components of the covariance. Specifically, when evaluating whether to select an observation $\psi_i^{\text{obs}}$ for carrying out computations at test input $x^*$ corresponding to some vertex $\nu$, we set the time component of $x_i^{\text{obs}}$ to that of the test input, $x_i^{\text{obs}\,\text{t}} \leftarrow x^{*\,\text{t}}$, making the temporal part of the covariance function irrelevant in this selection process. Observation selection is described in detail in Sect. 2.5.

2. Find a best-guess $\delta_\nu$ (and $\beta_\nu$, which is not used) by running the BFGS optimization algorithm (Nocedal, 1980) to find an approximate *maximum a posteriori* estimate by computing

$$
\begin{aligned}
\widetilde{\beta}_\nu, \widetilde{\delta}_\nu = \underset{\beta,\delta}{\arg\min} \bigg\{ & \sum_{j=1}^{n_\nu} (m(x_\nu; \beta_\nu, \delta_\nu) - \psi_\nu^{\text{obs}}{}_j)^2 \\
& + \sum_{\nu' \in \partial\nu} \big( (\delta_\nu - \delta_{\nu'})^T (\delta_\nu - \delta_{\nu'}) \\
& + (\beta_\nu - \beta_{\nu'})^T (\beta_\nu - \beta_{\nu'}) \big) \bigg\}.
\end{aligned}
\tag{12}
$$

The first sum runs over the training data selected by the observation selection method described in Sect. 2.5, and the second sum constrains the parameter values close to those in $\partial_\nu$. This optimization problem is very simple since there are few $\beta$ and/or $\delta$ parameters for the individual vertices.

3. Given $\widetilde{\delta}_\nu$, an estimate of the GP covariance parameters $\widetilde{\theta}$ – e.g. from a previous simulation or a best guess – and the observations $\psi_\nu^{\text{obs}}$, compute $\mathbb{E}[\beta_\nu | \psi_\nu^{\text{obs}}, \widetilde{\theta}, \widetilde{\delta}_\nu]$ and $\text{Cov}[\beta_\nu | \psi_\nu^{\text{obs}}, \widetilde{\theta}, \widetilde{\delta}_\nu]$ via Eq. (6) and (7). Together these give $p(\beta_\nu | \psi_\nu^{\text{obs}}, \widetilde{\theta}, \widetilde{\delta}_\nu)$.

If this computation uses a flat prior, as we do in this work, this is by Bayes' theorem proportional to the likelihood $p(\psi_\nu^{\mathrm{obs}}|\beta_\nu,\widetilde{\theta},\widetilde{\delta}_\nu)$.

4. Find the posterior marginal distribution of $\beta_\nu$ by applying Bayes' theorem and using the computed distributions at the neighboring nodes as prior information. Due to the Markov structure this becomes $p(\beta_\nu|\psi_\nu^{\mathrm{obs}},\widetilde{\theta},\widetilde{\delta}_\nu) \propto p(\psi_\nu^{\mathrm{obs}}|\beta_\nu,\widetilde{\theta},\widetilde{\delta}_\nu)\prod_{\nu^{ij}\in\partial_\nu} p(\beta^i$ If the spatial location corresponding to $\nu$ does not have any data to inform the fit (if $\psi_\nu^{\mathrm{obs}}$ is a zero-length vector), then parameter values from $\partial_\nu$ will determine the fit.

5. Using the $\beta_\nu$ obtained at the previous step, re-optimize *only* the $\delta_\nu$ parameters as above in step number 2. Since $\beta_\nu$ is not varied, the term $(\beta_\nu - \beta_{\nu'})^T(\beta_\nu - \beta_{\nu'})$ in Eq. (12) plays no role here.

The mean value of the distribution of $\beta_\nu$ coming out from step 4 corresponds to the $\hat\beta$ in e.g. Eq. 9, where $x^*$ would now refer to the spatial location of vertex $\nu$. Similarly, in case $\delta$-type coefficients are used, the functions $f_i$ will depend on the final $\delta_\nu$ values computed in step 5. The full sets of $\beta$ and $\delta$ coefficients for all the vertices in the graph are denoted by $\beta_\mathcal{V}$ and $\delta_\mathcal{V}$, and the sets of calibrated values are written $\hat\beta_\mathcal{V}$ and $\hat\delta_\mathcal{V}$.

## 2.4 Covariance functions in satGP

The smoothness, amplitude, and length scales of the Gaussian process realizations are determined by the covariance kernel used. The satGP program supports several different types of covariance function components for forming the full covariance function $k$ in Eq. (1). The options available reflect the properties that can be expected in remote sensing data – varying smoothness and meridional and zonal length scales, potential periodicity, and changing the orientation of the data-informed and uninformed axes according to wind speed and direction. This section lists the available covariance function formulations, and other forms may be easily added in the code.

For convenience, let

$$\xi_I^\gamma(x,x') \triangleq \sum_{c\in I}\left|\frac{x^c - x'^c}{\ell_c}\right|^\gamma = \|P^I(x) - P^I(x')\|_\Gamma^\gamma, \tag{13}$$

where $\gamma > 0$ is the exponent, $I$ is a (sub)set of the dimensions of the inputs $x$ and $x'$, $\ell_c$ are length scale parameters corresponding to the different dimensions in $I$, e.g. temporal ($\ell_\mathrm{t}$), zonal ($\ell_\mathrm{lon}$), or meridional ($\ell_\mathrm{lat}$) directions, and $x^c$ are different components of the inputs, e.g. $x^\mathrm{t}$, $x^\mathrm{lat}$, and $x^\mathrm{lon}$. The spatial length scale parameters $\ell_\mathrm{lat}$ and $\ell_\mathrm{lon}$ are in units of distance on the surface of the unit sphere, corresponding to radians at the equator. The $P^I$ matrix projects $x$ onto indices/dimensions in $I$, $\Gamma$ is a diagonal covariance matrix with diagonal elements $\ell_c^2$, and the notation $\|r\|_\Gamma$ stands for $\sqrt{r^T\Gamma^{-1}r}$, where $r$ is an arbitrary vector of the appropriate size. For remote sensing data used in this work, space-only variables form the set $I_S \triangleq \{\mathrm{lat},\mathrm{lon}\}$, and for spatial and temporal variables together the notation $I_{ST} \triangleq \{\mathrm{lat},\mathrm{lon},\mathrm{t}\}$ is used. Notation $\mathrm{lat}$ and $\mathrm{lon}$ refer to the spatial components of $x$, collectively earlier referred to as $x^\mathrm{s}$, and $\mathrm{t}$ refers to the temporal component. The form of $\xi$ in Eq. (13) implies that the different dimensions have separate length scale parameters $\ell$. The exponent $\gamma$ in $\xi$ is, however, shared between the dimensions. For the set of all $\ell$-parameters over a set $I'$ of dimensions we write $\ell_{I'}$. All the covariance functions below depend on a parameter $\tau$, square of which determines the maximum covariance that is attained when $x = x'$.

The exponential family of covariance functions with parameters $\theta_{\exp} \triangleq [\tau, \ell_{I_{ST}}, \gamma]^T$ is defined by the covariance function

$$k_{\exp}(x, x'; \theta_{\exp}) \triangleq \tau^2 \exp\left(-\xi_{I_{ST}}^{\gamma}(x, x')\right). \tag{14}$$

The exponent $\gamma$ controls the smoothness of the samples from the Gaussian process, with $\gamma = 2$ yielding infinitely differentiable realizations.

The Matérn family of covariance functions, with $\theta_M \triangleq [\tau, \ell_{I_{ST}}, \nu]^T$ is given by the covariance

$$k_M(x, x'; \theta_M) \triangleq \frac{\tau^2 s^{\nu}}{\Gamma(\nu)2^{\nu-1}} K_{\nu}(s), \tag{15}$$

where $s = 2\sqrt{\nu}\xi_{I_{ST}}^{1}(x, x')$ and $\nu$ controls the smoothness parameter usually denoted by $\alpha$ via $\alpha = \nu + \frac{q}{2}$. The function $K_{\nu}$ is the modified Bessel function of the second kind of order $\nu$. With $q = 1$, the value $\nu = \infty$ corresponds to the squared exponential kernel and $\nu = 0.5$ to the exponential kernel with $\gamma = 1$. Despite this similarity between the Matérn and exponential kernels,

the realizations of the random function from the processes with values $\frac{1}{2} < \nu < \infty$ do not correspond to those with the kernel $k_{\exp}$ with any value of $\gamma$.

A periodic kernel with $\theta_{\mathrm{per}} \triangleq [\tau, \ell_{I_S}, \ell_{\mathrm{per}}]^T$ is defined in satGP by

$$k_{\mathrm{per}}(x, x'; \theta_{\mathrm{per}}) \triangleq \tau^2 \exp\left(-\frac{2}{\ell_{\mathrm{per}}^2}\sin^2\left(\pi\left[\frac{x^{\mathrm{t}} - x'^{\mathrm{t}}}{\Delta_{\mathrm{period}}}\right]\right) - \xi_{I_S}^2(x, x')\right). \tag{16}$$

The parameter $\Delta_{\mathrm{period}}$ is the period length, which is assumed to be well known *a priori* and therefore is not among the

parameters that are calibrated. The second term in the exponent controls the spatial dependence via length-scale parameters in $\ell_{I_S}$, and $\ell_{\mathrm{per}}$ determines how far the temporal covariance extends, modulo $\Delta_{\mathrm{period}}$.

satGP contains an additional covariance function that utilizes local wind information when computing the covariances. The underlying rationale is that winds affect how quantities of interest such as gases in the atmosphere or algae blooms in surface water spread. For this reason, if wind data is available, it is natural to try to use it for inference with the Gaussian process. We

define the wind-informed covariance kernel with parameters $\theta_{\mathrm{w}} \triangleq [\tau, \ell, \ell_{\mathrm{t}}, \rho, w^*]^T$ by

$$k_{\mathrm{w}}(x, x'; \theta_{\mathrm{w}}) \triangleq k_{\exp}(x_{\mathrm{w}}, x_{\mathrm{w}}'; \tau, \{\ell_{\|}, \ell_{\perp}, \ell_{\mathrm{t}}\}, 2). \tag{17}$$

The parameter $\rho$ in $\theta_{\mathrm{w}}$ defines how strongly the magnitude of the wind vector at the test input, $w^* \triangleq [w_{\mathrm{lat}}^*, w_{\mathrm{lon}}^*]^T$ (the last parameter in $\theta_{\mathrm{w}}$), affects the shape of the covariance. The kernel itself is an exponential kernel, where the spatial components of the vectors $x$ and $x'$ are transformed by wind data, and where the covariance lengths are transformed by wind speed. A

spatio-temporal vector $x = [x^{\mathrm{lat}}, x^{\mathrm{lon}}, x^{\mathrm{t}}]$ is transformed by wind to the vector $x_{\mathrm{w}}$ in a new coordinate system according to

$$x_{\mathrm{w}} \triangleq \begin{pmatrix} (x^{\mathrm{s}} - x^{*\mathrm{s}})^T w^{\|} \\ (x^{\mathrm{s}} - x^{*\mathrm{s}})^T w^{\perp} \\ x^{\mathrm{t}} \end{pmatrix}, \tag{18}$$

where $x^{\mathrm{s}}$ and $x^{*\mathrm{s}}$ are the spatial components of vectors $x$ and $x^*$, and where $w^{\|}$ and $w^{\perp}$ are the unit vectors in the lat-lon coordinates along and perpendicular to wind direction at the test input $x^*$.

The spatial scaling ($\ell$) parameters for $k_\mathrm{w}$, corresponding now to the covariance scales parallel and perpendicular to the wind direction, are given by

$$\ell_\parallel \triangleq \ell \sqrt[4]{1 + |w^*|\rho}, \qquad \ell_\perp \triangleq \ell. \tag{19}$$

The parameter vector for the exponential kernel then becomes $\theta_\mathrm{exp} \leftarrow [\tau, \ell_\parallel, \ell_\perp, \ell_t, 2]^T$, where the last element denotes the exponent $\gamma$ used by the exponential kernel. A number of possible covariance ellipses resulting from the transformation procedure are shown in Fig. 3. Some data sets, like OCO-2, incorporate wind information, and satGP does have the capability of

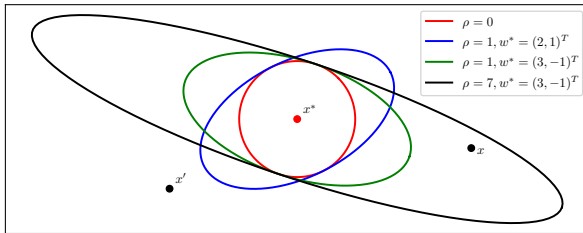

**Figure 3.** Equicovariance ellipses from the wind-informed kernel with various wind vectors $w^*$ and values of $\rho$. The wind values are taken at the test input $x^*$, but the covariance function $k$ is evaluated also for each pair of observations $x$ and $x'$.

gridding that data using another Gaussian process. Reading in gridded wind data from other sources is also a possibility. Using $k_\mathrm{w}$ requires that wind data at is available at each $x^*$.

The covariance functions used in this work to model $\Psi$ are sums of several kernels - sums of valid Gaussian process kernels remain valid kernels. The general form of the *multi-scale kernel* used in satGP is given by

$$k(x, x'; \theta) = \delta_{x,x'} \sigma_x^2 + \sum_{i=1}^{n_\mathrm{ker}} k_{\mathrm{ker}_i}(x, x'; \theta_{\mathrm{ker}_i}), \tag{20}$$

where the first term, which in kriging is called the nugget, contains the observation error variances, and where each $\mathrm{ker_i} \in \{\exp, \mathrm{M}, \mathrm{per}, \mathrm{w}\}$.

The kernel components of a multi-scale kernel are in this work called *subkernels*. The combined set of parameters is denoted by $\theta = [\theta_{\mathrm{ker}_1}^T, \ldots, \theta_{\mathrm{ker}_{n_\mathrm{ker}}}^T]^T$. Not all subkernel types are included in all experiments – rather, the simulations in Sect. 4 utilize kernels with one to three components. What those components should be depends on what fields are being modeled and what kinds of correlation structures the user expects to find in the data. Section 4.1 discusses identifiability of the different subkernel parameters of the multi-scale kernel.

Instead of calling $k(x, x'; \theta)$ in Eq. (20) a multi-scale kernel, the term multi-component kernel could also be used to describe the form. The term "multi-scale" underlines that the purpose of the combined kernel is to model well data, which contains several natural length scales, as remote sensing products often do. Furthermore, we believe that combining several kernels with identical length scale parameters does not represent a common use-case.

## 2.5 Covariance localization and observation selection for the multi-scale kernel

Using a large number of observations makes solving the Gaussian process Eq. (9) and (10) intractable as the cost of inverting the covariance matrix scales as $\mathcal{O}(n_{\mathrm{obs}}^3)$. This creates a need for finding approximate solutions while introducing as little error as possible. In satGP, covariance localization is used to utilize only a subset of observations for computing Eq. (9) and (10). To control the localization behavior the user needs to set two parameters: the maximum subkernel covariance matrix size $\kappa$ and the minimum covariance parameter $\sigma_{\min}^2$.

Assume that the multi-scale kernel defined by the user contains $n_{\mathrm{ker}}$ subkernels. For each test input $x^*$ and for each subkernel $k_l$ the set of observations feasible for inclusion in $K$ in Eq. (6) and (7) is

$$A_{*,l}^{\mathrm{obs}} \triangleq \{\psi_i \in \psi^{\mathrm{obs}} | k_l(x_i^{\mathrm{obs}}, x^*) > \sigma_{\min}^2, \psi_i \notin A_{*,j}^{\mathrm{obs}} \; \forall j < l\}, \tag{21}$$

where the last condition prevents observations from being added by several subkernels. In the end we select a single set of observations $A_*^{\mathrm{obs}}$ for each test input by combining some or all of the observations included in each $A_{*,l}^{\mathrm{obs}}$. The observation selection proceeds sequentially through the list of subkernels according to the procedure presented in Fig. 4. Recomputing

> **Data:** Set of feasible observations $A_{*,l}^{\mathrm{obs}}$ for each
> subkernel, maximum subkernel size $\kappa$,
> observation selection operator $\mathcal{S}$.
> **Result:** Set $A_*^{\mathrm{obs}}$ of observations that are
> informative for test input $x^*$
>
> 1   $A_*^{\mathrm{obs}} \leftarrow \emptyset$ ;
> 2   **for** $i \leftarrow 1$ **to** $n_{\mathrm{ker}}$ **do**
> 3      $\kappa' \leftarrow i\kappa - |A_*^{\mathrm{obs}}|$ ;
> 4      $A_*^{\mathrm{obs}} \leftarrow A_*^{\mathrm{obs}} \cup \mathcal{S}(A_{*,i}^{\mathrm{obs}}, \kappa')$;
> 5   **end**

**Figure 4.** Algorithm for selecting observations for carrying out predictions at test input $x^*$. The sets $A_*^{\mathrm{obs}}$ are defined by Eq. (21), and the variable $\kappa$ is the maximum subkernel size, also listed in Table 2 and discussed in Sect. 3. The selection operator $\mathcal{S}(A_{*,l}^{\mathrm{obs}}, \kappa')$ chooses $\kappa'$ observations from each $A_{*,l}^{\mathrm{obs}}$ either greedily or randomly.

the $\kappa'$ for each subkernel on line 3 of the algorithm allows selecting more than $\kappa$ observations by subkernels if the previous subkernels did not have $\kappa$ feasible observations available. This is done to allow the full kernel size to grow to $n_{\mathrm{ker}}\kappa$ when possible. On line 4, the observation selection operator $\mathcal{S}(A_{*,l}^{\mathrm{obs}}, \kappa')$ chooses $\kappa'$ observations from each $A_{*,l}^{\mathrm{obs}}$ either greedily by picking the observations with highest covariance with $x^*$, or randomly by sampling uniformly without replacement from $A_{*,l}^{\mathrm{obs}}$. Out of these two methods random selection avoids observation sorting and is therefore faster, especially if a huge number of data are near the test input $x^*$. This comes at the cost of producing noisier fields of marginal posterior means. For covariance parameter estimation random selection works well. See Appendix A for additional details.

Since the subkernels are handled sequentially, their order may affect which observations are selected due to the exclusion in Eq. (21), and to grow the full kernel to size $n_{\mathrm{ker}}\kappa$ as often as possible, it is recommended to specify the subkernel with

the largest $\ell$ parameters as the last one. After constructing $A_*^{\mathrm{obs}}$, the covariance matrix $K$ is constructed by evaluating the full covariance function $k$ according to Eq. (20) for all pairs of selected observations.

For learning the spatially varying $\beta$ and $\delta$ parameters for grid index $(i,j)$ in the mean function with the methods in Sect. 2.3.2, the observation selection is performed by disregarding the time component on the inputs, i.e. by setting $x_i^{\mathrm{obs}\,\mathrm{t}} \leftarrow x^{ij\,\mathrm{t}}$ for all $x_i^{\mathrm{obs}}$ in the training data. The reason for this is, that since learning the mean function amounts to fitting spatially varying parameter vectors $\beta$ and $\delta$, the data to perform the fit should not be selected based on covariance in the time direction as the mean function should be equally valid at all times.

Selecting the observations could be done also based on values of $k$ instead of each $k_l$ individually, or by other approaches, such as the one presented by Schäfer et al. (2017). However, even though the method of observation selection does have an effect on the inferred posterior marginals, the screening property of Gaussian processes ensures that this effect is not major as long as observational noise is small and the nearest observations are included in all directions. The parameter identifiability results in Sect. 4.1 and the WACCM4 results in Sect. 4.2 verify that the current nearest-neighbor-in-covariance approach works as intended.

## 2.6  Learning the covariance parameters $\theta$

From Sect. 2.1 the log marginal likelihood of observations $\psi^{\mathrm{obs}}$ given a set of parameters $\theta$, $\beta$ and $\delta$ is given by

$$2\log p(\psi^{\mathrm{obs}}|\beta,\delta,\theta) = -\|(\psi^{\mathrm{obs}} - F\beta)\|_K^2 - \log|K| - n_{\mathrm{obs}}\log(2\pi), \tag{22}$$

where the covariance function parameters $\theta$ implicitly determine $K$, and the non-linear space-dependent mean function parameters $\delta$ affect the values in $F$. The maximum (marginal) likelihood estimate (MLE) $\hat{\theta}$ of $\theta$ can be found via minimizing

$$\mathcal{L}(\theta) = \|(\psi^{\mathrm{obs}} - F\hat{\beta})\|_K^2 + \log|K| + n_{\mathrm{obs}}\log(2\pi) \tag{23}$$

as stated in context of Eq. (5).

In the presence of a huge number of observations, calculating the determinant of the full covariance $|K|$ is not feasible, and maximizing the log likelihood is approximated by

$$\hat{\theta} = \underset{\theta}{\arg\min} \sum_{x_i^* \in E_{\mathrm{ref}}} \left\{ \|(\psi^{\mathrm{obs}}_{\mathrm{local}_i} - F_i\hat{\beta})\|_{\widetilde{K}_i}^2 + \log|\widetilde{K}_i| \right\}, \tag{24}$$

where $E_{\mathrm{ref}}$ is a set of randomly sampled points from the spatio-temporal domain specified for the experiment, determined by the parameters area and $n_{\mathrm{days}}$ in Table 2. The $\hat{\beta}$ and $\hat{\delta}$ parameters, the latter of which is embedded in $F$, are the point estimates corresponding to each $x_{i*}$, interpolated from the values obtained for the full grid. The optimization in Eq. (24) is carried out over all subkernel parameters with some caveats: currently the smoothness-related parameter $\nu$ for the Matérn kernel, and the exponent $\gamma$ for the exponential kernel are not calibrated, and naturally neither are the wind data $w^*$ listed as a parameter for the wind-informed covariance – however, the parameter $\rho$ affecting that kernel can be learned.

While the selection of inputs included in $E_{\mathrm{ref}}$ has an effect on the obtained parameter estimate, that effect has proven in simulations to be small. The vectors $\psi^{\mathrm{obs}}_{\mathrm{local}_i} \in \mathbb{R}^{d_i}$, where $d_i$ is the number of observations chosen by the observation selection

method of Sect. 2.5 for test input $x_i^*$, contain observations closest in covariance to $x_i^*$, each of which is a reference point included in $E_{\text{ref}}$. The matrices $F_i$ are the corresponding $F$-matrices, as described in Sect. 2.1. The last term in Eq. (23) is dropped, since while varying $\theta$ in Eq. (24) changes $d_i$, the number of total observations in the problem should fundamentally stay the same.

The maximum likelihood estimate approximation in Eq. (24) contains a sum over blocks of observations, which can together be thought of as a block-diagonal approximation of the full dense covariance for all observations in all $\psi_{\text{local}_i}^{\text{obs}}$. The blocks in this approximation are the dense covariance matrices $\widetilde{K}_i$, and in contrast to a full dense $K$, in this approximation the cross-covariances between observations in $\psi_{\text{local}_i}^{\text{obs}}$ and $\psi_{\text{local}_j}^{\text{obs}}$, $i \neq j$, are set to zero. This is done even if the randomly selected corresponding inputs $x_i^*$ and $x_j^*$ are close to each other. Due to the $\mathcal{O}(n^3)$ cost of inverting the covariance matrix, which is

needed for finding the maximum likelihood estimate, using the block approximation provides a critical efficiency improvement without which learning the covariance function parameters would not be feasible.

While this method is suitable for finding point estimates for the parameters $\theta$, the computed approximated log-likelihood has an unknown scaling factor resulting in an unknown multiplicative factor for the variance term in the exponent of the Gaussian distribution, and hence information about the true size of the posterior distribution of the covariance parameters

$p(\theta|\psi^{\text{obs}}, \hat{\beta}_{\mathcal{V}}, \hat{\delta}_{\mathcal{V}})$ is lost.

By default the scaled posterior $p(\theta|\psi^{\text{obs}}, \hat{\beta}_{\mathcal{V}}, \hat{\delta}_{\mathcal{V}})$ is explored by using the Adaptive Metropolis (AM) Markov chain Monte Carlo (MCMC) algorithm (Haario et al., 2001), an implementation of which is included in the satGP source code. MCMC methods (Gamerman, 1997) are used to draw samples from probability distributions when direct sampling is not possible, but the likelihood function can still be evaluated. The samples are drawn by generating a Markov chain of parameter values, which

is an autocorrelated sample from the posterior. The AM algorithm is an adaptive method that is efficient for many real-world sampling situations. The observation selection procedure in Sect. 2.5 introduces discontinuities to the posterior distribution due to selected observations changing when the covariance function parameters are modified. Computing $\hat{\theta} \leftarrow \mathbb{E}[\theta|\psi^{\text{obs}}, \hat{\beta}_{\mathcal{V}}, \hat{\delta}_{\mathcal{V}}]$ with MCMC — i.e. using the posterior mean of a Monte Carlo sample — usually works around this noisiness in the likelihood. On the downside, MCMC is computationally much more demanding than finding the maximum a posteriori estimate with

optimization, since MCMC may require computing up to millions of likelihood evaluations. In the satGP context using MCMC is feasible since the forward model simply amounts to sampling from a multivariate normal distribution which is very fast. Furthermore, the parameter dimension is moderate, even with multiple subkernels, limiting the need to generate extremely long chains. The current version of satGP uses a flat prior distribution for the covariance parameters, with hard limits on the parameter ranges.

The software also includes a capability to learn the covariance parameters using optimization algorithms such as COBYLA or SBPLEX available in NLOpt. These methods are much faster than MCMC, but have the tendency of getting stuck in local minima, limiting their usefulness.

## 3 Overview of Computation

The satGP code is written in C with visualization scripts written in Python and parallelization implemented with OpenMP directives. The program reads data from netCDF and text files and the configuration from a C header file. For linear algebra satGP uses the C interfaces of LAPACK and BLAS, LAPACKE and CBLAS, and optimization tasks are carried out with the NLOpt library. The computations are performed in single precision in order to save memory resources with the largest data sets, and also to improve performance.

The most important configuration variables are listed in Table 2. The user needs to define whether parameters are learned or

**Table 2.** Most important satGP control variables and high level C structs: first section contains parameters for program logic, second for domain specification, third for covariance and mean function definition, and last for observation handling. This list is by no means exhaustive – the configuration file contains lots of variables that can control the program. Some additional tweaking is possible by changing hard-coded values directly in the source code, such as those listed in Appendix A.

| Variable | Type | Low | High | Notes |
|---|---|---|---|---|
| learn_k | `int` | 0 | 2 | (0) Don't train $\theta$, (1) generate observations and learn $\theta$, (2) learn $\theta$ from non-synthetic data. |
| learn_m | `int` | 0 | 1 | (0) Don't train local $\beta$ and $\delta$, (1) find local $\beta$ and $\delta$ as in Sect. 2.3. |
| sampling | `int` | 0 | 2 | (0) Skip sampling, (1) calculate GP marginals at each grid point, (2) sample from GP. |
| area | `char*` | - | - | Area definition setting longitude and latitude minimum and maximum values |
| $n_{\text{days}}$ | `int` | 1 | $\infty$ | Number of days to be simulated |
| $\omega$ | `float` | $> 0$ | 180 | 1-d grid resolution in degrees – small values degrade esp. posterior sampling performance. |
| $n_{\text{ker}}$ | `int` | 1 | 10 | Number of subkernels $k_l$ in $k$ |
| cfc | `struct*` | - | - | Recursive struct pointer defining $k_1 \ldots k_{n_{\text{ker}}}$ and corresponding $\theta$, see Sect. 2.4. |
| mf | `struct*` | - | - | Struct pointer for defining type of $m(\cdot,\cdot)$ and associated (initial) $\beta$ and $\delta$, see Sect. 2.2. |
| $\zeta_{\text{train}}$ | `float` | 0 | $\infty$ | Fraction of observations that are randomly included in $\psi^{\text{obs}}$ when learning $\theta$, $\beta$, and $\delta$. |
| $\zeta_{\text{sample}}$ | `float` | 0 | $\infty$ | Fraction of observations that are randomly included in $\psi^{\text{obs}}$ when sampling $\neq 0$. |
| $\sigma^2_{\text{min}}$ | `float` | 0 | $\infty$ | Discard observation at $x_i^{\text{obs}}$ for $x^*$ if $k(x_i^{\text{obs}}, x^*) < \sigma^2_{\text{min}}$, see Sect. 2.5. |
| $n_{\text{ref}}$ | `int` | 0 | $\infty$ | Number of reference points in $E_{\text{ref}}$ in Eq. (24) for training $\theta$ |
| $n_{\text{synthetic}}$ | `int` | 0 | $\infty$ | Number of random locations where synthetic data is generated for training $\theta$ |
| $\sigma^2_{\text{synthetic}}$ | `float` | 0 | $\infty$ | Variance of Gaussian noise added to synthetic observations |
| $\kappa$ | `int` | 1 | $\infty$ | Maximum subkernel size, values $\kappa > n_{\text{ker}}^{-1} 1000$ will be slow due to $\mathcal{O}(\kappa^3)$ scaling. |

prescribed and whether marginals or samples from the GP are to be computed. The mean function and the covariance kernel are defined by initializing corresponding structs with parameters and their limits if calibration is to be performed. For computing GP marginals or drawing samples from the random process, the geographic and temporal extents need to be specified and the mean function and the covariance kernel used must be given.

Several parameters can be tweaked to improve computational efficiency, including all of those in the second and last sections of Table 2. The first main bottleneck for computing a marginal at $x^*$ is sorting the observations for selecting the most informative ones to be used in the covariance matrices, see Sect. 2.5. This requires roughly $\mathcal{O}(r_l \log r_l + \kappa \log \kappa)$ operations for each subkernel, where $r_l$ is the number of grid locations $x^{ij}$ in the spatio-temporal grid such that for the $l^{\text{th}}$ subkernel, $k_l(x^{ij}, x^*) > \sigma_{\min}^2$. For subkernels with $\gamma = 2$, $r_l \propto \prod_{i=1}^{q} \ell_i^l$, with $\ell_i^l$ denoting the length scale parameters over all the dimensions of the inputs $x$. In other words, $r_l$ is proportional the size of the hypersphere inside which observations are considered for each $x^*$. The second bottleneck is calculating the Cholesky decompositions of the covariance matrices $K$ with cost $\mathcal{O}((n_{\text{ker}}\kappa)^3)$. The cost of calculating the means and variances of the GP in a grid for a set of $n_{\text{times}}$ points on the time axis is therefore given by

$$\text{cost} = \mathcal{O}\left(\frac{An_{\text{times}}}{\omega^2}\left[(n_{\text{ker}}\kappa)^3 + \sum_{l=1}^{n_{\text{ker}}}(r_l \log r_l + \kappa \log \kappa)\right]\right), \tag{25}$$

where $A$ is the grid area in degrees squared and $\omega$ is the grid resolution. When the random observation selection method mentioned in Sect. 2.5 is used, the $r_l \log r$ in Eq. (25) becomes just $r_l$.

The execution of the program is presented in Fig. 5. The function `AddToState()` reads observations (asynchronously) into a **state** object that tracks the proximity of each observation to each grid point. Only a part of the observations is added, controlled on line 6 by the parameter $\eta_{\text{train}}^i$, which corresponds to the inclusion probability of each observation. This probability depends on $\zeta_{\text{train}}$ in Table 2 via

$$\eta_{\text{train}}^i \triangleq \frac{d(x_i^{\text{obs}}, x_{i_{\text{prev}}}^{\text{obs}})}{\omega \zeta_{\text{train}}} \wedge 1, \tag{26}$$

where $d(x_i^{\text{obs}}, x_{i_{\text{prev}}}^{\text{obs}})$ is the Euclidean distance of the point at $x_i^{\text{obs}}$ that is being proposed for addition to the previous added point at $x_{\text{prev}_i}^{\text{obs}}$ and $\wedge$ is the standard notation for minimum. Hence with $\zeta = 0$ all observations are added.

For computing the marginals, the spatial domain can be decomposed with `Decompose()`, line 23, into several spatial subdomains (sd) so that arbitrary-size grids can be computed. This makes solving large problems with limited amount of memory possible, but only works with **sampling**$= 2$. This option is in practice rarely needed, and it was not needed for the simulations in Sect. 4. The state object is emptied by `ReInitializeState()` which also potentially sets new subdomain extents. Function `SampleFromPrior()` actually performs the computations on lines 30-37, but with the inputs $x^*$ in a random pattern instead of in a grid as is the case in l. 27-38.

The `AddSubdomainData()` method on l. 29 adds data as on lines 3-9, but only to the current subdomain. After that, the `SelectObservations()` method (l. 31) carries out selecting the best observations as described in Sect. 2.5. For constructing the set of potential observations, the grid is searched for locations that may have informative observations for the current test input stored in the **state** object. These locations are first ordered into categories with decreasing potential covariance and for the best locations, that together hold at least $2\kappa$ observations, the covariance function with the test input is evaluated. Out of these, the $\kappa$ best are chosen. The factor 2 can be increased for the wind-informed kernel and the value 8 is used in the demonstration in Sect. 4.8.

**Data:** filelist containing files with observation data
$y_i = (\mu_{\psi_i}, \sigma^2_{\psi_i})$ indexed by location $x_i$, input
variables from Table 2.

**Result:** Optimized $\beta$ parameters for mean
function and $\theta$ parameters for covariance
kernel, gridded Gaussian process marginal
means and variances or a sample from the
Gaussian process evaluated in a grid.

*Initialization: Create grid according to* area *and* $\omega$,
  *define* $k(x,x')$ *and* $m(x,t)$, *initialize* state;
**if** learn_m $= 1$ **or** learn_k $= 2$ **then**
**for** file **in** filelist **do**
D $\leftarrow$ ReadData (file);
**for** $(x_i, y_i) \in$ D **do**
**if** Bernoulli($\eta^i_{\text{train}}$) **then**
AddToState(state, $x_i, y_i$);
**end**
**end**
**end**
**if** learn_m **then** FindLocalMeanfunCoeffs (state);
**if** learn_k $= 1$ **then**
ReInitializeState (state, fulldomain);
**for** $i \leftarrow 1$ **to** $n_{\text{synthetic}}$ **do**
$(x_i, y_i) \leftarrow$ SampleFromPrior ( );
AddToState(state, $x_i, y_i$);
**end**
**end**
**if** learn_k $\neq 0$ **then**
FindCovfunCoeffs ($n_{\text{ref}}$)
**end**
**if not** sampling **then**
$(\mathsf{n}_{\text{sd}}, (\mathsf{sd}_i)^{\mathsf{n}_{\text{sd}}}_{i=1}) \leftarrow$ Decompose($\mathsf{n}^{\max}_{\text{dom}}$, area, $\omega$);
**else**
assert ($\mathsf{n}_{\text{gp}} < \mathsf{n}^{\max}_{\text{dom}}$);
**end**
**if** sampling **then for** $i \leftarrow 1$ **to** $\mathsf{n}_{\text{sd}}$ **do**
ReInitializeState (state, $\mathsf{sd}_i$);
AddSubdomainData (state, filelist, $\mathsf{sd}_i$, $\eta_{\text{sample}}$);
**for** $x^* \in \mathsf{sd}_i$ **do**
$A^{\text{obs}}_* \leftarrow$ SelectObservations(state, $x^*$);
$\mu^*, \sigma^2_* \leftarrow$ ComputeMarginal($x^*$, $A^{\text{obs}}_*$);
**if** sampling $= 2$ **then**
$\widehat{\psi^*} \leftarrow$ Normal($\mu^*, \sigma^2_*$);
AddToState(state, $x^*$,($\widehat{\psi^*}, \sigma^2_{\text{synthetic}}$))
**end**
**end**
**end**

**Figure 5.** Overview of satGP execution. After initialization data is read for training $m$ and $k$ and possible MRF computation is carried out. This is followed by sampling the prior if a synthetic study is performed, and learning the $\theta$ parameters controlling $k$. Gaussian process marginals are then computed in a grid, potentially by decomposing the domain for large grids. Finally, samples from the GP may be drawn. The names of the subprograms here deviate from those in the code to improve readability.

The function `ComputeMarginal()` constructs the covariance matrix $K$, inverts via the Cholesky decomposition, and solves Eq. (9) and (10) to find the marginal distribution at any test input $x^*$. That function returns the negative log likelihood and is therefore directly used in learning the covariance parameters $\theta$ in `FindCovfunCoeffs()` on line 18.

The Gaussian process algorithm is an interpolation algorithm when observation noise is zero, and interpolation algorithms may misbehave when used for extrapolation. In a spatio-temporal large grid, when sampling $= 2$, i.e. when draws of the Gaussian process are generated in a regular spatio-temporal grid, computing conditionals based on the previous predictions would amount to extrapolation if done in order. For this reason, a deterministic sparse ordering is used, which ensures that test inputs corresponding to simultaneous predictions are far from each other so that their mutual covariance is negligible. Conditioning on already computed values is therefore for the vast majority of GP evaluations interpolation instead of extrapolation.

## 4    Results and discussion

In this section we present several simulation studies. The first experiment examines parameter identifiability with the multi-scale kernel using satGP-generated data. We then demonstrate how satGP posterior distributions look like compared to truth using synthetic ozone fields from the WACCM4 model.

After that we concentrate on analyzing satGP results produced using the OCO-2 Level 2 data. First, we learn the parameters of the locally varying mean function of the form in Eq. (2) by computing the MRF, and those fields are then analyzed. We then learn the covariance parameters of the OCO-2 XCO2 spatio-temporal field from data. Knowing both the mean and the covariance functions allows us to evaluate the Gaussian process globally in a grid and we present snapshots of the global mean and uncertainty fields. The section concludes by comparing posterior marginal fields generated by using single-scale and multi-scale kernels and by demonstrating how the wind-informed kernel works.

### 4.1    Parameter identifiability with the multi-scale kernel

We performed a synthetic study to confirm the identifiability of the multi-scale covariance function parameters. The synthetic data was generated by satGP by sampling from zero-mean processes with known covariance parameters and with a random spatial pattern from the prior, adding 1% noise. The parameters were then estimated by computing the posterior mean estimates using Adaptive Metropolis.

The identifiability experiment was performed with various kernels, and recovering the true parameters was the more difficult the more complex the kernel was. With a single Matérn, exponential, or periodic kernel, the parameters could be recovered very easily. This was also true for a combination of exponential and Matérn kernels with a relatively small $\kappa$ parameter.

The covariance kernel parameters were still recoverable with a combination of three kernels, Matérn with $\nu = \frac{5}{2}$, exponential, and periodic. This setup required using a larger $\kappa = 256$. With small $\kappa$, some of the parameters had a tendency to end up at the lower boundary, possibly due to effects of the covariance cutoff on the determinant of the covariance matrix in Eq. (22). Optimization using minimization algorithms such as Nelder-Mead, COBYLA, or BOBYQA tended to often end up in local

minima, and for this reason MCMC was used instead. The number of random reference points in $E_{\mathrm{ref}}$ in Eq. (24) was set to 12, which was enough to reliably recover parameters close to the true value.

The parameter limits, true values, and posterior means of the synthetic experiment with three kernels are given in Table 3. In total 200,000 observations were created in the region between -10 and 10 latitude and -10 and 10 longitude over a period
of four years according to the true values reported in Table 3. A total of 10 million MCMC iterations were computed to make sure that the posterior covariance stabilized. The posterior, with first 50% of the chain discarded as burn-in, is shown in Fig. 6

**Table 3.** Lower and upper limits, with true and estimated parameter values. The three-kernel synthetic covariance function parameter estimation problem is already very difficult, here resulting in slight overestimation of the parameters of the smallest kernel.

|  | low | high | true | est | $\frac{est-true}{true}$ |
|---|---|---|---|---|---|
| $\tau^{\mathrm{mat}}$ | 0.05 | 1 | 0.5 | 0.652 | 0.304 |
| $\ell_{\mathrm{lat}}^{\mathrm{mat}}$ | 0.003 | 0.02 | 0.007 | 0.00989 | 0.413 |
| $\ell_{\mathrm{lon}}^{\mathrm{mat}}$ | 0.003 | 0.02 | 0.01 | 0.0135 | 0.350 |
| $\ell_{\mathrm{t}}^{\mathrm{mat}}$ | 1d | 14d | 7d | 8.06d | 0.15 |
| $\tau^{\mathrm{per}}$ | 0.01 | 2 | 1 | 1.073 | 0.073 |
| $\ell_{\mathrm{lat}}^{\mathrm{per}}$ | 0.001 | 0.04 | 0.02 | 0.0207 | 0.035 |
| $\ell_{\mathrm{lon}}^{\mathrm{per}}$ | 0.001 | 0.04 | 0.02 | 0.0220 | 0.1 |
| $\ell_{\mathrm{per}}$ | 0.01 | 0.3 | 0.1 | 0.1075 | 0.075 |
| $\tau^{\mathrm{exp}}$ | 0.5 | 3 | 1 | 0.927 | -0.077 |
| $\ell_{\mathrm{lat}}^{\mathrm{exp}}$ | 0.005 | 0.1 | 0.025 | 0.0352 | 0.408 |
| $\ell_{\mathrm{lon}}^{\mathrm{exp}}$ | 0.005 | 0.1 | 0.04 | 0.0405 | 0.0125 |
| $\ell_{\mathrm{t}}^{\mathrm{exp}}$ | 7d | 30d | 21d | 24.83d | 0.182 |

How well parameters can be learned from data depends always on the data and the exact Gaussian process form chosen. While the identifiability studies presented here show that the parameter calibration procedure works and that covariance parameters are recoverable in a synthetic settings, identifiability cannot be always expected. Still, even in these cases, the MAP
and/or posterior mean estimates of the covariance parameters should provide good point estimates for $\theta$.

## 4.2    Posterior predictive distribution from synthetic WACCM4 ozone data

A synthetic study using WACCM4-generated ozone data was conducted to verify and to illustrate that the methods to learn the model parameters $\beta$, $\delta$, and $\theta$ produce a realistic GP regression model that then produces credible posterior predictive fields. In a synthetic setting the mean values of the posterior predictive distributions should be close to the true fields, and the
discrepancies between the ground truth and the predicted fields need to be explainable by the predicted marginal uncertainties. The role of this part in the study is to give an example of how a Gaussian process predictive posterior field produced with satGP compares with the underlying true field.

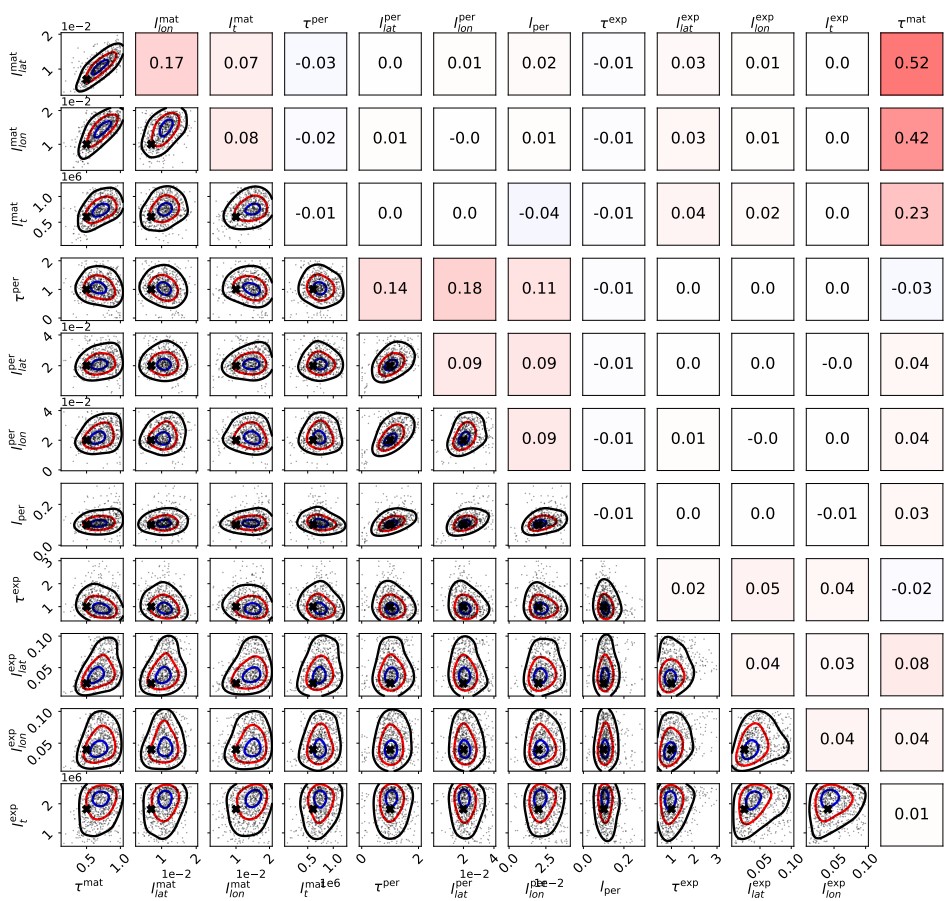

**Figure 6.** Scaled MCMC posteriors from a synthetic study where data was generated with a multi-scale Gaussian process. The figure demonstrates that even with three subkernels, multi-scale Gaussian process kernel parameters can be recovered. The lower left part shows the pairwise marginal distributions of the parameters, and the black crosses denote the true parameter values. The axis labels are on the left and below the figure. The upper right triangle shows sample correlations between the parameters from the chain, with axis labels on the left and on the top. Small within-subkernel positive correlations are present. The contours shown include 85% (black), 50% (red), and 15% (blue) of the posterior mass.

The WACCM4 model is an atmospheric component of the Community Earth System Model from NCAR (Hurrell et al., 2013), capable of comprehensively representing atmospheric chemistry and modeling the atmosphere up to thermosphere. WACCM4-generated ozone data for the years 2002-2003, with a latitude-longitude grid resolution of $1.9° \times 2.5°$, 88 vertical levels going up to roughly 140 km, and with an internal time step of 30 min, were used as ground truth and to generate synthetic observations. Since the model was used for generating synthetic two-dimensional data, a specific atmospheric sigma hybrid pressure level of $3.7$ kPa was selected.

Ozone data at approximately 400 locations were sampled daily over a two-year period in a random pattern from the domain of the experiment to learn the parameters of the mean and covariance functions. The training data set was then generated by interpolating to these points from the simulated WACCM4 data. This sampling procedure corresponds to creating on average one observation daily for each $12.5° \times 12.5°$ longitude-latitude square.

Using these data, the mean function parameters were fitted locally using the method in Sect. 2.3.1, utilizing the functions $f_i$ in Eq. (11), but with two additional terms $f_5$ and $f_6$, which were similar to the $f_1$ and $f_2$ except for different $\Delta_{\mathrm{period}}$ parameters and phase shift parameters $\delta$, that were shared between these $f_5$ and $f_6$ only. These functions were used to model periodical behavior with 2 and 1.5 year period lengths. The covariance function parameters of a kernel consisting of a single Matérn kernel, Eq. (15), were learned using the approximate maximum likelihood technique described in Sect. 2.6 with data from the first year. The parameter $\nu$ used for the kernel was $\frac{5}{2}$. The optimization was carried out with MCMC and the posterior mean estimate of the covariance parameters was selected for $\hat{\theta}$. The values of the covariance parameters obtained were $\tau = 0.589$, $\ell_{\mathrm{lat}} = 0.143$, $\ell_{\mathrm{lon}} = 0.225$, and $\ell_t = 2$ d $16$ h $15$ min. That $\ell_{\mathrm{lon}}$ is larger than $\ell_{\mathrm{lat}}$ echoes the OCO-2 results presented later in Table 4.

For computing the posterior predictive distributions, the observational data $\psi^{\mathrm{obs}}$ were sampled from the WACCM4 simulations at locations closest in space and time to where the GOMOS instrument made measurements during its first year of operation. No noise was added to these observations. The posterior predictive distribution was computed for one full year, and he total number of observations used was 39538. The reason for using different spatial patterns for learning the model parameters and for running the Gaussian process regression was that with this choice, the quality of the fit of the mean and covariance functions was not dependent on the spatial location, and therefore, if major spatial discrepancies between the ground truth and the posterior predictive fields had emerged, those could then have been attributed to the GOMOS sampling pattern used to generate the synthetic observations $\psi^{\mathrm{obs}}$.

The marginal posterior predictive distributions were computed globally in a uniform grid with the resolution of $2.5°$ in east-west direction and $1.9°$ in the north-south direction between $78.63°$S and $78.63°$N and daily over the period from Jan 6 2002 to Jan 5 2003, totaling around 4.384 million marginals in the predictive posterior. The one year long computation took 19 min 18 s on a relatively fast Intel i7-8850H laptop CPU.

Figure 7 shows the ground truth from WACCM4 with the mean field and corresponding marginal uncertainties obtained from satGP for Dec. 2 2002. The ground truth and the estimated fields are very similar, and the uncertainty is higher when there are no observations nearby. The posterior mean field retains a lot of fine detail from the ground truth and is not overly smoothed or sharp, suggesting that the covariance parameter calibration procedure has found a well-performing estimate for

the covariance parameters $\theta$. The smallest reported uncertainties are close to zero, as they should, due to lack of observation error.

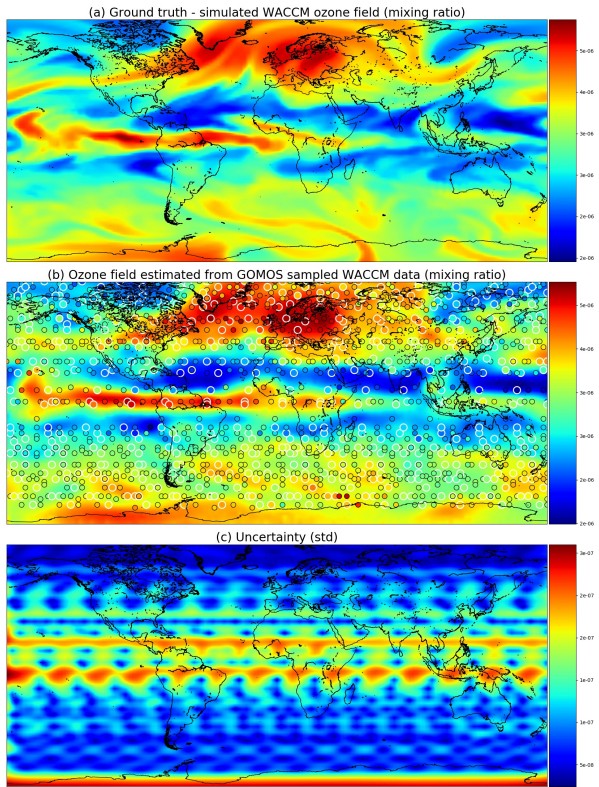

**Figure 7.** Ozone field mixing ratios at $3.7$ k Pa for Dec. 2 2002. Panel (a) shows the simulated ground truth from WACCM4, (b) is the GP posterior mean, and (c) gives the posterior predictive uncertainties. A single Matérn kernel was used. In (b) the larger circles with the white edges are observations from Dec. 2, and the smaller circles stand for observations from Dec. 1 and Dec. 3.

## 4.3 The OCO-2 v9 data

The simulations with non-synthetic remote sensing data use the v9 data from the OCO-2 satellite. OCO-2 was launched in 2014, and it orbits the Earth on a Sun-synchronous orbit (Crisp et al., 2012; O'Dell et al., 2012), completing 14.57 revolutions around Earth in one day. The footprint area of each measurement is roughly $1.29 \times 2.25$ k m$^2$, but the data is very sparse in time and in space. In the presence of clouds, the satellite is not able to produce measurements, and this poses a challenge for areas with persistent cloud covers, such as Northern Europe in the winter.

The present work uses the XCO2 data, its reported uncertainties, associated coordinate information, and zonal and meridional wind speeds that are contained in the data files. The time period considered is from Sept. 6 2014 to Nov. 10 2018 and we use only observations flagged as good, of which there are in total 116489342.

## 4.4 Solving the mean function for OCO-2 v9

Calibrating the mean function from OCO-2 v9 XCO2 data as described in Sect. 2.3.1 produces the estimates for the $\beta$ and $\delta$ coefficients shown in Fig. 8. The $\beta_i$ parameters are the coefficients of the functions $f_i$ in Eq. (11), and $\delta$ is the phase shift parameter in $f_1$ and $f_2$. The upper left quadrant of Fig. 8 shows the semiannual variability of the XCO2 concentration. The timing of winter and summer in the Northern and Southern hemispheres explains the color shift along the equator. The lower left quadrant shows the amplitude of the twice faster oscillations, and like $\beta_1$, also $\beta_2$ shows the highest amplitude oscillations in the boreal region.

The constant term $\beta_3$ in the upper right quadrant shows the background concentration. Some of the reddest areas such as East China, both coasts of the United States, Central Europe, and the Persian Gulf stand out, and they are also areas where major emission sources are known to exist. Finding local elevated concentrations compared to surrounding areas echoes the observations made by Hakkarainen et al. (2016), where empirically defined time-integrated local XCO2 anomalies were interpreted as possible emission sources. The trend component $\beta_4$ varies only a little spatially, due to CO2 mixing in the atmosphere over time, and for this reason it is not shown here. The phase shift parameter $\delta$ is modeled separately, and the field in the lower right quadrant is obtained by optimization, conditioning on the $\beta$ factors. This partly explains the different spatial pattern. The figure shows how the phases of the XCO2 annual cycles differ between regions, but the $\delta$ values need to be interpreted together with the $\beta_1$ and $\beta_2$ coefficients.

At high latitudes XCO2 observations from OCO-2 are available only for a short period every year, and the quality of these measurements is often poorer. For this reason the calibration procedure may yield unrealistic and noisy values close to the poles, especially for parameters $\beta_1$ and $\beta_2$. The obtained parameter values closer than $20°$ to the northern and southern edges of the domain were averaged by setting the parameter values at each $x^{ij}$ to $\beta^{ij} \leftarrow \frac{\hat{\beta}^{ij}d}{20} + \frac{(20-d)\overline{\beta}}{20}$, where $d$ is the distance to the edge of the domain in degrees, $\hat{\beta}^{ij}$ is the calibrated parameter vector at $x^{ij}$, and $\overline{\beta}$ is the average value of the parameters over the area where $x^{ij}$ is located and where averaging is performed. The $\delta$ parameter was treated similarly. This adjustment was done as a post-processing step after finding the mean function coefficients. The main benefit of performing this adjustment is, that the posterior predictive distributions become more realistic in winter at high latitudes when the mean function dominates.

Figure 1 shows time series of the mean function for a variety of locations, verifying that the exact form chosen is able to describe much of the local variability in XCO2.

## 4.5 Covariance parameters of the OCO-2 v9 data

The OCO-2 data has several natural spatial and temporal length scales. The distance between adjacent observations is only one to two kilometers in space and some hundredths of a second in time, but the distance between consecutive orbits is thousands of kilometers in space and several hours in time. On consecutive days the satellite passes close to the trajectory of the previous day at a distance of tens to three hundred kilometers depending on the latitude. The Earth has natural temporal diurnal and annual cycles, but since OCO-2 is Sun-synchronous, only the latter matters with OCO-2 data. Since the annual cycle is already fitted by finding the mean function coefficients $\beta_1$ and $\beta_2$ in Eq. (11) corresponding to the periodic functions $f_1$ and $f_2$, a periodic

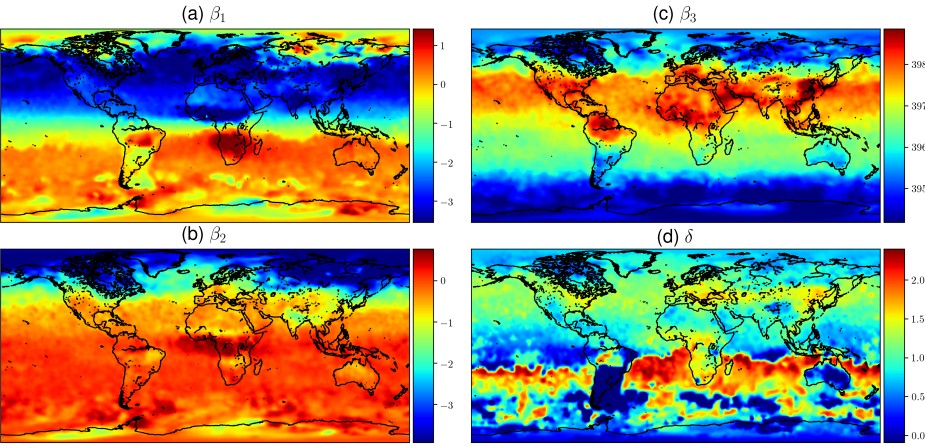

**Figure 8.** Mean values of mean function coefficients that were described as a Markov random field, calculated in a $2° \times 2°$ grid between $85°$ N and $85°$ S. The $\beta_i$ coefficients multiply the $f_i$ functions in Eq. (11). Panel (d) shows how the phase parameter $\delta$ can vary more in the southern hemisphere where $\beta_1$ and $\beta_2$ are small. The mean function and fitted daily means for several locations with the corresponding mean function parameters are shown in Fig. 1.

covariance kernel component is not included. The OCO-2 data is therefore modeled with a kernel consisting of a larger-scale exponential and a smaller scale Matérn subkernel.

The covariance parameters for the two-component kernel are given in Table 4. The values used were the median values from sampling the posterior with MCMC. When learning the parameters from a data set with several natural length scales, the posterior may appear multi-modal, with some of the modes only having relatively little mass. In such a case, the median provides a more robust estimate for the parameters than the mean. The $\ell_{\mathrm{lon}}^{(\cdot)}$ and $\ell_{\mathrm{t}}^{(\cdot)}$ parameters of the posterior mean were slightly larger, which would result in slower computation. Selecting the median is further justified by the slight overestimation of some parameters in the synthetic study in Sect. 4.1.

Learning the covariance parameters from OCO-2 v9 data used the following configuration parameters for satGP: $\zeta_{\mathrm{train}} = 0$, $\kappa = 256$, and $n_{\mathrm{ref}} = 12$. A total of 1.1184 million MCMC iterations were completed, with the first 50% discarded as burn-in to produce statistics. The reference points were randomly picked from a rectangle with corners at ($0°$S, $65°$E) and ($60°$N, $145°$E). While using the whole globe would have been a principled choice, MCMC requires lots of iterations, and for any claim of global coverage $n_{\mathrm{ref}}$ would have needed to be much larger.

### 4.6 Posterior predictive distributions of XCO2 from the OCO-2 v9 data

The marginal posterior predictive distribution at test points $x^*$, given by Eq. (9) and (10), were calculated globally in a half-degree grid between $80°$S and $80°$N at daily time resolution. The first day of simulation was September 6 2014, and the last day was November 10 2018, spanning in total 1526 days. For each day, 230400 marginals were computed, resulting in a collective 351 million inverted covariance matrices. The satGP parameters used were $\zeta_{\mathrm{sample}} = 0$ and $\kappa = 256$, and the covariance kernel

**Table 4.** Covariance function parameter values learned from OCO-2 data. First column shows the Matérn kernel parameters, and the second column the exponential kernel parameters. The spatial length scale parameters are given as distance on the unit sphere, with $0.01$ corresponding to approximately $64\,\mathrm{k\,m}$. The length scales along the parallels, $\ell_{\mathrm{lon}}^{(\cdot)}$ are much larger than that along the meridians, $\ell_{\mathrm{lat}}^{(\cdot)}$.

|  | $(\cdot) = \mathrm{mat}$ | $(\cdot) = \mathrm{exp}$ |
|---|---|---|
| $\tau^{(\cdot)}$ | 0.899 | 2.72 |
| $\ell_{\mathrm{lat}}^{(\cdot)}$ | 0.00513 | 0.0418 |
| $\ell_{\mathrm{lon}}^{(\cdot)}$ | 0.0363 | 0.397 |
| $\ell_{\mathrm{t}}^{(\cdot)}$ | 20h 22min | 16d 20h 12min |

used was the one learned in Sect. 4.5, with parameters given in Table 4. The simulation wall time was 26 days on a moderately fast Intel i7-8700K CPU utilizing the available 12 CPU threads and 32 GiB memory.

Figures 9 and 10 present global fields of the mean values and marginal uncertainties, with a subset (to avoid excessive over-drawing) of observations shown as a scatter plot in the (a) panels. The (b) panels show how uncertainty is reduced with the overpass of OCO-2. This uncertainty reduction diminishes fast due to the Matérn component of the multi-scale kernel having a very short length scale parameter in the time dimension. In the upper figures the background color (mean of the Gaussian process posterior) usually matches the observations, but due to observational noise, the posterior mean is not everywhere an interpolated field.

### 4.7 Comparison of single- and multi-scale kernels with OCO-2 data

Data from the OCO-2 can be used to demonstrate how the multi-scale kernel formulation affects the predictive posterior distributions. Figure 11 shows posterior marginals from September 15 2014. The first row (a-b) contains results from the multi-scale kernel described in Sect. 4.5, and the second row (c-d) shows fields from only the exponential part of the multi-scale kernel. The parameters of the multi-scale kernel are shown in Table 4. The bottom row (e-f) contains the difference fields between the first and the second rows. The single-kernel uncertainty is very low in Fig. 11 (d) since lots of observations fall into regions of high covariance with almost any test input, with the exception of the northern side of Ireland, which does not have any observations nearby. Since the covariance kernel parameters were trained for the multi-scale kernel, the parameters used for the single kernel are not the ones describing the XCO2 field best.

Panel (a) shows that as intended, the multi-scale approach leads to local enhancements of the XCO2 mean field. Far from the measurements, the smaller Matérn kernel no longer reduces the predicted marginal uncertainties, and this leads to an increase in uncertainty in these areas. Figure (e) shows additional enhancements of the XCO2 mean fields, which are in this case due to the different maximum covariances between the multi-scale and single-scale kernels.

The total kernel size was kept at 1024 ($\kappa = 512$ for (a-b) and $\kappa = 1024$ for (c-d)) in both experiments, and thinning and grid resolution parameter values were $\zeta_{\mathrm{sample}} = 5$, and $\omega = 0.5°$. The very same observations were used for both simulations.

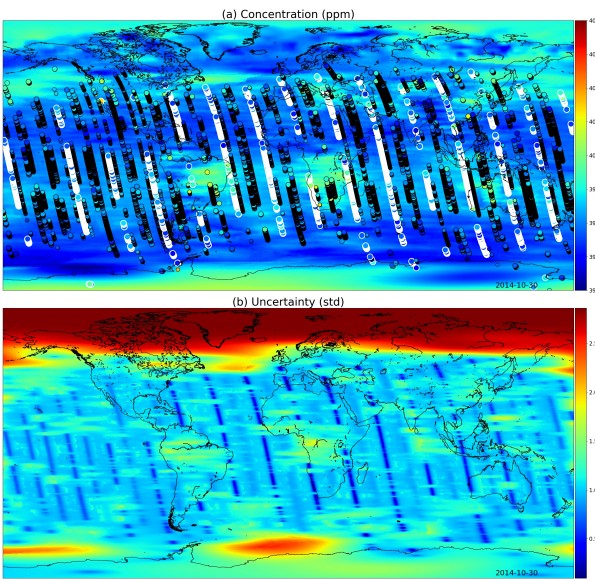

**Figure 9.** XCO2 posterior mean values (a) and their uncertainties (b) on the $30^{\text{th}}$ of October 2014. The most informative observations are shown with the concentrations, with the large white circles being from the $30^{\text{th}}$, medium circles from one day before or after, and small circles from two days before or after. The OCO-2 utilizes sunlight for retrieval, which is why there are very few observations above $60°N$. These fields include latitudes up to $85°S$ and $85°N$.

## 4.8 Wind-informed kernel with OCO-2 data

The wind-informed kernel, Eq. (17), lets local wind data at test input $x^*$ rotate and scale the axes along which the covariance between two points is computed. Modeled winds are included with OCO-2 data, and they can be used to produce gridded winds that can then be used locally with the computation of each marginal posterior predictive distribution.

The covariance parameters for a single wind kernel were learned by taking the median of an MCMC posterior, similarly as was done in Sect. 4.5. The resulting parameters were $\tau = 2.07$, $\ell = 0.038$, and $\rho = 56.7$. The variance of $\rho$ was high, possibly due to the square root in the current formulation in Eq. (19). For this simulation, $\zeta = 1$, $\kappa = 1024$, and $\omega = 0.7$, and the simulation time for the area from $(27°N, 115°E)$ to $(40°N, 145°E)$ for the single day was 2.652s (walltime) on the i7-8750H laptop CPU.

The simulation results are shown in Fig. 12. Low uncertainties shown in blue color on the right spread with the winds, as do the concentration estimates on the left both due to the high reading in South Korea and the low reading close to Shanghai.

     Optimally the wind-informed kernel should utilize winds that are not recomputed from the observations as was done here for convenience, but directly from a weather or climate model or from a wind data product. The satGP program contains configuration options for doing this. The optimal covariance function parameter values are conditional on the wind data, so the

values should be learned separately for each new application and wind data set.

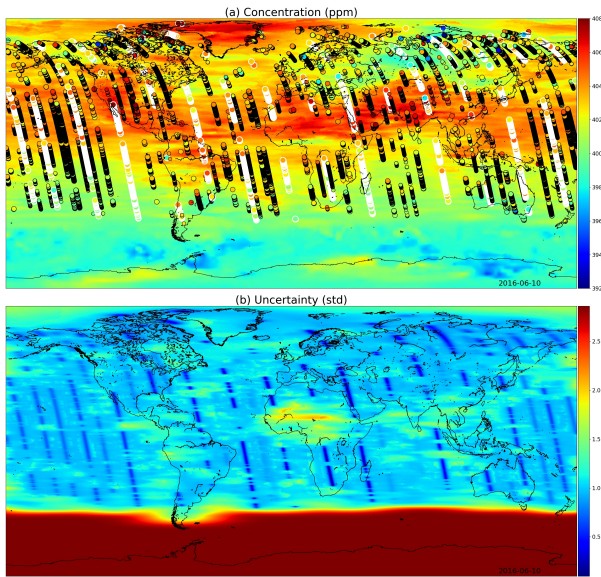

**Figure 10.** XCO2 posterior mean values (a) and their uncertainties (b) on $10^{\text{th}}$ of June 2016. While photosynthesis in the Northern Hemisphere is already reducing the carbon dioxide concentrations globally, the observations condition the Gaussian process to higher mean values than in Fig. 9. In the summer months the uncertainty stays high close to the South Pole. These fields include latitudes up to $85^{\circ}$S and $85^{\circ}$N.

## 5 Conclusions and future work

In this work we introduced the first version of a fast general purpose Gaussian process software, satGP v0.1.2, which is in particular intended to be used with remote sensing data. We showed how the program solves spatial statistics problems of enormous sizes by using a spatially varying mean function, learned by computing marginals of an MRF, and by using a multi-
scale covariance function, parameters of which are found either by using optimization algorithms or with adaptive Markov chain Monte Carlo. We also presented how satGP allows conducting synthetic parameter identification studies by sampling from Gaussian process prior and posterior distributions, and this could be done with any kernel prescribed, including a non-stationary wind-informed kernel. The features of satGP were demonstrated first with a small scale synthetic ozone study, and then using the enormous XCO2 data set produced by the NASA Orbiting Carbon Observatory 2.

Various aspects of satGP can be improved in future versions, some of which include improving the observation selection/thinning scheme for statistical optimality, adding support for multivariate models and higher input dimensions, and adding methods for finding locally stationary model parameters to be able to describe heterogeneous scenes better. Despite all the room for development, satGP is a useful tool already in its present state, and it may with little additional modeling be used e.g. to fuse data from different sources, such as GOSAT, GOSAT-2, OCO-2, TANSAT, and OCO-3. This will enable producing
more precise posterior estimates, and with that a more complete picture of the evolution of for instance the atmospheric carbon dioxide distribution. Such statistically principled products that incorporate uncertainty information can then be used as a robust backbone for both making policy decisions and further scientific analyses.

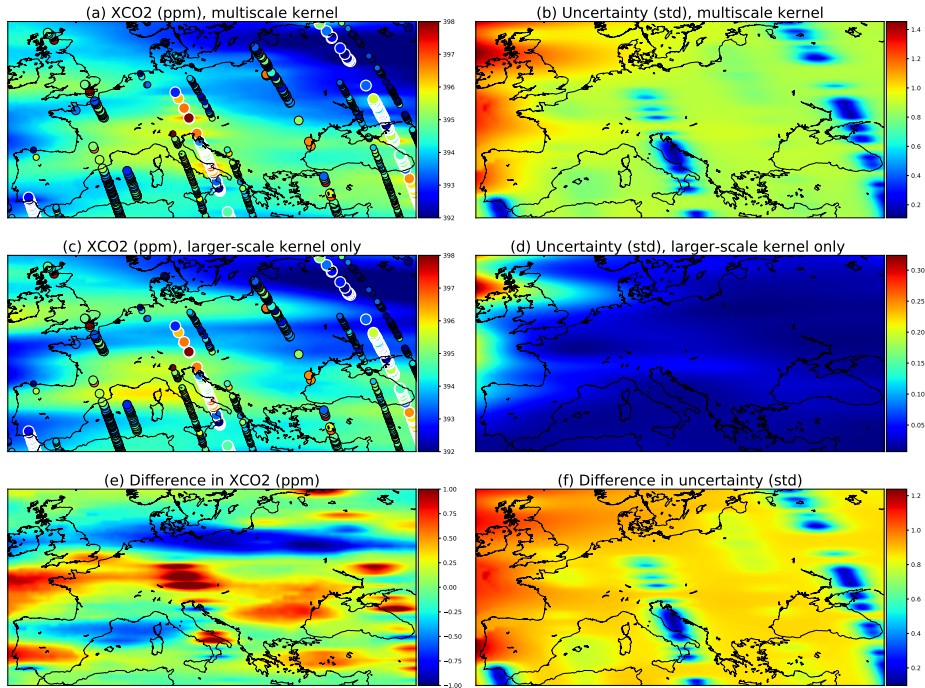

**Figure 11.** Comparison of a multi-scale kernel with the two components described in Sect. 4.5 and a single component kernel defined by the parameters of the exponential kernel. These parameters were given in Table 4. The observations used are the same and are shown in panels (a) and (c) as circles. The large ones with white borders are observations from the present day, September 15 2014, medium circles are observations from $14^{th}$ and $16^{th}$, and small circles from $13^{th}$ and $17^{th}$.

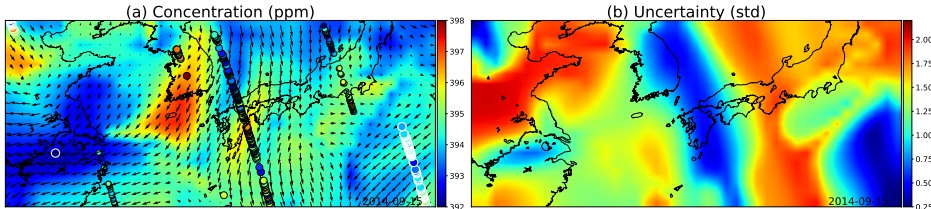

**Figure 12.** (a) GP posterior mean of XCO2 and (b) its uncertainties with the wind-informed kernel. The area shown contains the Korean peninsula in the center, China on the left, and Japan on the center-right. The large circles with the white edges are present-day observations, medium circles are observations from adjacent days, and the smallest ones are observations from two days away. Wind direction and magnitude are given by the black arrows, and uncertainty is clearly reduced where wind is blowing directly towards or away from the observations.

*Code and data availability.* The satGP code is available as a supplemnt to this paper under the MIT license. The OCO-2 v9 data used is freely available directly from NASA. The WACCM4 model is available from UCAR as a component of the Community Earth System Model.

## Appendix A: Input parameters and variables in satGP

The satGP software by design allows for a lot of flexibility for defining how to model the quantity of interest as a Gaussian random field. This section goes over those possibilities and some practical recommendations. The parameters in Table 2 are described in more detail than earlier, along with some other configuration variables in the configuration file `config.h`. Some of the details in this section may change for future versions of the software.

Of the four sections in Table 2, the first is obvious, as those parameters control the main logic of satGP. It is recommended to first learn the mean function, then with that mean function learn the covariance function, and only after that calculate the means and variances of the Gaussian process with sampling $= 1$. The setting sampling $= 2$ can be used e.g. for illustration purposes, to understanding how the different realizations of the random function would look like, or to generate synthetic data products.

The area parameter defines the longitude-latitude extents of the domain where satGP performs the computations. The strings and the corresponding areas are defined in the beginning of the file `gaussian_proc.h`, and can be changed there as needed. Current available areas contain e.g. `NorthAmerica`, `Europe`, `EastAsia`, `World`, and `TESTAREA`.

The parameter $n_{\mathrm{days}}$ defines how many days are to be simulated after the starting day. Currently the starting day is hard coded in the code to be the first day of OCO-2 data. However, if `use_daylist` $\neq 0$ in the configuration file, a list of days can be used. This list can quite easily generated by modifying a trivial python script `create_daylist.py`, which is included with satGP.

The $\omega$ parameter determines how much spatial detail is resolved when sampling or computing marginals of the random field. A small value like 0.1 will make computing very expensive, and using such values might be unnecessary when the smallest covariance subkernel length scale parameters are large. The spatial $\ell$ parameters are in the scale of distances on the unit ball, and therefore on the equator an $\ell$ parameter of 0.05 corresponds to a length scale of around $2.9°$, so the $\omega$ parameter should rarely be much less than half of that. On the other hand, if the observations are spatially very close to each other and describing local variation is aimed for, then the $\ell$ parameters also need to be small. Given computational constraints, larger values or different area parameters may need to be used.

In the third section of Table 2, the first parameter $n_{\mathrm{ker}}$ denotes the number of subkernels. Even though the hard limit is set to 10, in practice this should be between one and three since the parameters of more than three subkernels are not necessarily reasonably identifiable. More kernels means more computational cost, due to the $\kappa$ parameter, which is the last one in the table and is discussed later.

The parameters cfc and mf are not strictly input variables, but C struct pointers that are created based on input variables. These variables are described in the configuration file, and they amount to choosing the covariance kernels from prescribed types (e.g. Matérn, exponential, and periodic), and then defining the parameters for those kernels. The best parameters are those that are learned with learn_k $= 2$ when non-synthetic data is used.

Learning the covariance parameters $\theta$ is best performed with MCMC, and the posterior mean and median have proven to be useful values. For unimodal posterior distributions these values are usually very close to each other. The number of MCMC it-

erations is controlled by the variable `mcmc_iters`, for which $10^6$ is a large enough value. The number of reference points $n_{\mathrm{ref}}$ in the set $E_{\mathrm{ref}}$ in Eq. (24) that is used for computing the log-likelihood can be set to a low value of e.g. number of CPU threads, if at least 12 are available. If with MCMC the chain gets stuck in local minima, the value of the `mcmcs->scalefactor` in the `mcmc()` function in `mcmc.h` may be shrunk, and equally well, if the posterior ends up being flat with respect to many parameters, it may be increased. This is justified since due to the approximate maximum likelihood method correct scaling factor of the log posterior density is in any case unknown.

For learning the covariance parameters, parameter limits need to be given. These should correspond to the expected length scales in the data – e.g. long-range fluctuations with low amplitude, and short-scale variations due to local effects. It is in practice best if the parameter ranges do not overlap.

If the exponent of the exponential kernel needs to be changed, that needs to be done by changing the `exponent` variable in the `covfun_dyn()` function in the file `covariance_functions.h`. Similarly, if the order of the Matérn kernel needs to be changed, that can be done by changing the variable `n` in functions `covfun_matern52()` and `initialize_covfunconfig()` in that same file.

For constructing the mean function, the configuration file contains the parameter `mftype`. The possible values are: 0) a zero mean function is used, 1) a mean function that changes only in time is used, 2) a (time-dependent) field is read in and used - this can be e.g. the mean value from a previous Gaussian process simulation, and 3) a space and time dependent mean function is used. The function itself is given as a function pointer to variable `mean_function` in the configuration file, and this function needs to be defined somewhere – e.g. in the file `mean_functions.h`. For the mean function, another variable, `mfcoeff`, needs to be set. This is the total number of parameters ($\beta$ and $\delta$ in Eq. (2)) if `mftype` $\in \{1,3\}$. If the mean function parameters are learned, the parameter `nnonbetas`, the number of mean function non-linear $\delta$ parameters, needs to be set to the appropriate value in the function `fit_beta_parameters_with_unc()` in `mean_functions.h`. For global mean function coefficients, the values of those coefficients are given in the configuration file, where the parameter limits for learning the space-dependent mean function parameters are also set. Finally, when learning the space-dependent mean function parameters, the smoothness of the field may be controlled by changing the `dscale` parameter in the configuration file, and to a lesser extent by modifying the `dfmin` and `dfmax` parameters in function `fit_beta_parameters_with_unc()` in file `mean_functions.h`. Another strategy for e.g. producing smoother mean function coefficient fields is to use high values for $\zeta_{\mathrm{train}}$ and $\kappa$ and large spatial length scale parameters in the covariance kernel. Changing the priors for the $\beta$ parameters is done in section 2 of `fit_beta_parameters_with_unc()` in `mean_functions.h`.

In the last section, the $\zeta_{\mathrm{train}}$ parameter controls data thinning when learning covariance kernel parameters and the $\zeta_{\mathrm{sample}}$ does the same when sampling $\neq 0$. How the thinning takes place was explained in the context of Eq. (26). While with few observations no thinning needs to be done at all, i.e. $\zeta$ may be set to zero, with large data sets the representability of data may be improved when a coarse grid is used for computation, and also memory bottlenecks may be avoided. These parameters may be increased if faster execution is required, for example for debugging purposes.

The $\sigma_{\min}^2$ parameter controls which observations are not considered at all when computing at a location $x^*$, as described by Eq. (21). The higher this is, the more data is discarded. Setting $\sigma_{\min}^2$ to a very low value makes searching for candidate

observations slow, while picking too high a value may make posterior fields look edgy. In practice values between $10^{-7}$ and $10^{-3}$ seem to work well. This parameter is not meant to be changed often, due to which it is set in `create_config()` in the file `gaussian_proc.h`.

The variable $n_{\text{synthetic}}$ defines how many synthetic observations are generated when learn_k $= 1$. Very large values are once again expensive, and instead a smaller area should rather be used with more moderate values of $n_{\text{synthetic}}$. Those values can be in practice up to $10^5$ or more. With very low values, it may be that spatial patterns specified by the prescribed covariance kernel are not represented appropriately, and therefore values less than $10^4$ should be avoided, except for maybe in setups with only a single subkernel. If $\sigma^2_{\text{synthetic}}$ is high, parameter identifiability suffers. What values are enough large also depends on the maximum covariance parameters of the Gaussian process, given by the $\tau^2$ parameters in the formulas of Sect. 2.4.

The last parameter in Table 2, $\kappa$, defines the maximum subkernel size. The larger this parameter is, the more data is included for constructing the covariance matrix $K$, whose Cholesky decomposition needs to be computed to solve the local regression problem inherent to Gaussian processes. In practice the full kernel size should be kept under 1000, and in order to compute GP calculations fast, a full kernel size of less than 500 is recommended. However, with a very small number of marginals, values up to $10^4$ may be experimented with. When $n_{\text{ker}}\kappa < 64$, the speed-up due to solving the GP formulas faster decreases, since at that point computing Cholesky decompositions no longer takes up the majority of the computing time. This lower bound depends on the CPU architecture and the sizes of the various CPU caches.

Whether the observations for computing the local values are chosen at random or greedily is determined by the variable `select_closest` in function `pick_observations()` in file `covariance_functions.h`. The value used should normally be non-zero, since with random selection adjacent grid points often do not utilize the best available observations closest by, leading to noisiness or graininess in the posterior mean field.

In addition to the parameters and variables listed here, there are also other parameters in the configuration file and in the code, even though those should not need to be changed. Any variables that the user might want to tweak are generally accompanied by at least some comments describing their effects.

In the current version, the satGP program is run with the script `gproc.sh`, whose comments describe the various options. Compiling and running require a modern GCC version (such as version 8) and the meson build system, and additionally all the needed libraries listed in Sect. 3. The current low version number reflects the fact that as of now, installing and using the software will require a degree of technical knowledge, including some Python, C, and BASH programming skills.

*Author contributions.* JS, AS, HH, and YM designed the study. TH produced the WACCM4-specific results. JS prepared this manuscript, wrote the satGP code, chose, tested and implemented the computational methods, and performed the non-WACCM4 simulations, with contributions from all co-authors.

*Competing interests.* The authors declare that they have no conflict of interest.

*Acknowledgements.* This work was supported by the Centre of Excellence of Inverse Modelling and Imaging (CoE), Academy of Finland, decision number 312122. We would like to thank Pekka Verronen and Monika Andersson from the Finnish Meteorological Institute for providing the WACCM4 data fields. The research was partly carried out at the Jet Propulsion Laboratory, California Institute of Technology, under a contract with the National Aeronautics and Space Administration (80NM0018D0004). Copyright 2020. All rights reserved. US Government Support Acknowledged.

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
