# Peer review of "Efficient multi-scale Gaussian process regression for massive remote sensing data with satGP v0.1.2"

_Geoscientific Model Development, 2019_

## Referee Comment (RC1) · Anonymous Referee #1 · 24 Sep 2019

This manuscript describes a model to analyze large spatio-temporal data. Although analyzing remote sensing data of enormous sizes is no double important and challenge, the manuscript fails to describe the model and its computation details and properties sufficiently or clearly. Please see below my comments that are not necessarily ordered chronologically or by importance:

1. This manuscript suggest using the mean function of a particular form when analyzing OCO-2 data: $m(x; \beta, \delta) = \tilde{f}(x^t; \delta(x^s))'\beta(x^s)$ as given in Equation (3). This mean function is not a linear form of unknown parameters $\{\delta(x^s), \beta(x^s)\}$, noting that they are both dependent (i.e., varying) across locations. I find the description

on how to estimate $\delta(x^s)$ and $\beta(x^s)$ extremely confusing.

- In Lines 10-20 of Page 6, it states that $\beta$ will be estimated using the formula of generalized least squared as given in Equation (6), and $\delta$ will be calibrated, but no explanation is given on how $\delta$ will be *calibrated*. In addition, the authors did not explain the dimension of the matrices $F$ and $K$ in Equation (6). Are they large so that $K^{-1}$ or $(F^T K^{-1} F)^{-1}$ difficult to compute?

- How is $\beta(x^s)$ estimated for a location $x^s$? For a location $x^{s,test}$ *without* data/observation, can we estimate $\beta(x^{s,test})$ and how?

- Although the authors have included Section 2.4 on learning $\beta(x^s)$ as a Markov random field, this section is not connected to other parts of the manuscript but only adds confusion. It is unclear what the authors meant by modeling $\beta(x^s)$ as a Marko random field. Does this mean that the authors no longer use Equation (6) to estimate $\beta(x^s)$? What are the assumptions of this Markov random field (MRF)? What are the parameters in this MRK and how is this MRK fitted?

- It is also confusing how the parameters $\delta(x^s)$ are estimated.

- Line 14 of Page 8: " ... finding $\hat{\beta}$ with Eq. (9) and (10), ..." Is this a typo? Should it be Eq. (6) and (7)?

- Page 8 Line 15: The objective function $\sum_{j=1}^{n}(m(x_\nu; \beta_\nu, \delta_\nu) - \psi_j)^2 + \sum_{j' \in \partial\nu}(\delta_\nu - \delta_{j'})^2$ and the optimization procedure are poorly explained. It should be noted that the mean function $m(\cdot; \cdot, \cdot)$ involved $\delta$ and $\beta$. It is very confusing how or why this function is used to estimate $\delta$ or $\beta$ individually or both of them jointly, and why it should be used this way.

2. The notations in this manuscript are very confusing overall. For example, the authors sometimes use $\beta(^s)$ and later use $\beta_\nu$. The covariance parameters are even more confusing. There are $l$, $l_c$, and $l_I$. Even the definition of $I$ is not consistent: It is originally stated $I \subset \{x^s, x^t\}$, but later used as $I = ST$ or $I_S$,

and $I_{ST}$. Also, the authors used $\Delta_{year}$ in Equation (11) and stated $\Delta_{year}$ is the duration of one year, does this mean $\Delta_{year} = 365$? Similarly, in Equation (15), the authors used $\Delta_{period}$; is it 365 as well?

3. The authors suggest the multi-scale covariance function given in Equation (18): $k(x, x'; \theta) = \delta_{x,x'} \sigma_x^2 + k_{per}(x, x'; \theta, I_S) + k_M(x, x'; \theta) + k_{exp}(x, x'; \theta, I + ST) + k_W(x, x'; \theta, I)$.

- First, I am not sure multi-scale is an accurate way to describe this covariance function. I feel this function is to add different *types* of covariance functions together, but these components not necessarily differ in terms of scales.

- The authors did not explain clearly the component $k_W(x, x'; \theta, I)$. Although Equation (16) states it is equal to $k_{exp}(x_W, x'_W; \theta^W, ST)$, the authors fail to explain $x_W$ or the quantifies in Equation (17) especially, $l$, $l_t$, $l^{\|}$, and $l^{\perp}$, and how these parameters are chosen/estimated.

- What will happen if there are missing data in wind velocity?

- Why isn't there an $I$ involved in the Matérn component $k_M(\cdot, \cdot; \cdot)$?

- For the exponential component, the definition given in Equations (12) and (13) are not clear. At least there are two ways to define this component:

$$k_{exp}(x, x'; \theta, I_{ST}) = \tau^2 exp(-|\frac{x - x'}{l_{ST}}|^{\gamma})$$

or

$$k_{exp}(x, x'; \theta, I_{ST}) = \tau^2 exp(-|\frac{x^s - x^{s'}}{l_S}|^{\gamma^s}) exp(-|\frac{x^t - x^{t'}}{l_T}|^{\gamma^t})$$

dependent on whether the spatial and temporal components share the scale or exponent parameters. I don't know what the authors have used, and there is no justification of their choice.

- The authors need to provide a better description of these components in the covariance function and explain why they are identifiable based on their formulations and definitions. Also, it is necessary to clarify whether some parameters are the same or vary across these components, such as $\tau^2$, $\gamma$, and $l$.

4. I find Sections 2.6 and 2.7 quite difficult to understand. It seems that the authors use local kriging, that is, using a subset of data close to a prediction location $x^*$ to estimate the covariance parameters and to make prediction. Furthermore, it appears that the authors use different subsets of data to estimate the components in the covariance function. Why not using a single subset data to estimate the entire covariance function? Or, were the authors trying to avoid identifiability issue by using different data sets to estimate different covariance components? If a subset of data are used, I assume the size of this chosen subset is not too large, but why is there a need to use a block diagonal matrix $\tilde{K}_i$ as in Equation (22)? This approximation is not clearly explained, neither is $E_{ref}$ in Equation (22). Moreover, in Equation (19), should it be $> \sigma^2_{min}$ rather than $< \sigma^2_{min}$?

5. The authors mentioned the nearest neighbor Gaussian process, but did not cite the reference correspondingly.

6. It is unclear where or why MCMC is needed and how it is implemented (prior specification etc.). The authors described optimization in Section 2 and also in the first paragraph of Page13. However, later in Page 17, the authors stated that MCMC is used instead. Section 2 does not describe MCMC.

7. It should be Matérn covariance function, instead of Matern.

---

## Referee Comment (RC2) · Anonymous Referee #2 · 1 Oct 2019

Review of "Efficient multi-scale Gaussian process regression for massive remote sensing data with satGP v0.1"

by Jouni Susiluoto, Alessio Spantini, Heikki Haario, and Youssef Marzouk

The manuscript considers the important and challenging problem of modelling global $CO_2$ satellite data with Gaussian processes. The problem is computationally extremely difficult, especially as the authors utilise laptop-based solutions instead of running the model on supercomputing facilities. The authors set a consistent probabilistic framework within Bayesian statistical estimation theory, and fully describe their model. The key features of the benefits include: computationally fast (C/LAPACK/BLAS), spatiotemporal multiscale kernels, spatial dependence of the mean function estimated as a Gaussian Markov random field, can do large-scale observations and marginals (e.g. 10ˆ8), covariance localisation, methods to estimate mean and covariance function parameters.

In general, solving general spatial statistics problems with Gaussian processes, one gets means, uncertainties, and realisations of GPs. Paper contributes for space-dependent mean function estimation, multiscale kernel for many spatial scales, data-driven covariance parameter estimation, and the final result is computation of a posteriori distributions for OCO-2 datasets.

The manuscript is timely, well-written and makes a significant contribution to GPs and geophysics. Hence, the paper should be published without any delays.

---

## Short Comment (SC1) · 5 Nov 2019

Dear authors,

in my role as Executive editor of GMD, I would like to bring to your attention our Editorial version 1.2:

https://www.geosci-model-dev.net/12/2215/2019/

This highlights some requirements of papers published in GMD, which is also available on the GMD website in the 'Manuscript Types' section:

http://www.geoscientific-model-development.net/submission/manuscript_types.html

[Figure]

In particular, please note that for your paper, the following requirements have not been met in the Discussions paper:

- "Code must be published on a persistent public archive with a unique identifier for the exact model version described in the paper or uploaded to the supplement, unless this is impossible for reasons beyond the control of authors. All papers must include a section, at the end of the paper, entitled "Code availability". Here, either instructions for obtaining the code, or the reasons why the code is not available should be clearly stated. It is preferred for the code to be uploaded as a supplement or to be made available at a data repository with an associated DOI (digital object identifier) for the exact model version described in the paper. Alternatively, for established models, there may be an existing means of accessing the code through a particular system. In this case, there must exist a means of permanently accessing the precise model version described in the paper. In some cases, authors may prefer to put models on their own website, or to act as a point of contact for obtaining the code. Given the impermanence of websites and email addresses, this is not encouraged, and authors should consider improving the availability with a more permanent arrangement. Making code available through personal websites or via email contact to the authors is not sufficient. After the paper is accepted the model archive should be updated to include a link to the GMD paper."

Therefore please provide the satGP v0.1 code or provide the reasons why the code can not be made publicly available in your revised submission to GMD.

Yours,

Astrid Kerkweg

---

## Author Comment (AC1) · 12 Feb 2020

We thank the anonymous reviewers and the editor for carefully reading the manuscript and for providing the very valuable comments. We address the comments one by one below. The reviewer comments are pasted verbatim below in italics, and the author responses to these comments can be found immediately under the comments, starting "A:".

**Anonymous Referee #1**

*This manuscript describes a model to analyze large spatio-temporal data. Although analyzing remote sensing data of enormous sizes is no double important and challenge, the manuscript fails to describe the model and its computation details and properties sufficiently or clearly. Please see below my comments that are not necessarily ordered chronologically or by importance:*

1. *This manuscript suggest using the mean function of a particular form when analyzing OCO-2 data: $m(x; \beta, \delta) = f(x^t; \delta(x^s))\beta(x^s)$ This mean function is not a linear form of unknown parameters $\{\delta(x^s), \beta(x^s)\}$, noting that they are both dependent (i.e., varying) across locations. I find the description on how to estimate $\delta(x^s)$ and $\beta(x^s)$ extremely confusing.*

- *In Lines 10-20 of Page 6, it states that $\beta$ will be estimated using the formula of generalized least squared as given in Equation (6), and $\delta$ will be calibrated, but no explanation is given on how $\delta$ will be calibrated. In addition, the authors did not explain the dimension of the matrices $F$ and $K$ in Equation (6). Are they large so that $K^{-1}$ or $(F^T K^{-1} F)^{-1}$ difficult to compute?*

A1: First, we mention that we find a point estimate for the $\delta$ parameters before calibrating $\beta$ with generalized least squares, and that we then still one more time calibrate the $\delta$ parameters. We agree that the wording could be better, and we will clarify the alternating optimization in the sentence under (7) for the revised manuscript, adding that we use optimization algorithms for the task. We also give a reference to a later section for the full description of the procedure. Second, the reviewer is absolutely correct about that the matrices describing the joint probability density of all the $\beta$ coefficients are too large for direct inversion. In the OCO-2 simulations the size of $K$ is up to the order of $10^8 \times 10^8$. We will clarify the sizes of these matrices in the text.

- *How is $\beta(x^s)$ estimated for a location $x^s$? For a location $x^s$, test without data/observation, can we estimate $\beta(x^s, test)$ and how?*

A2: The $\beta^s$ is estimated via the Markov Random Field, by fitting the parameters to match the mean function to local observations, and by conditioning on the parameter values at neighboring spatial locations. When there is no data nearby, the values of the parameters will be determined by prior values (if any – we use a flat prior) and the parameters at neighboring nodes in the MRF. We agree that the description in Section 2.4 is at the moment not very clear, and we will describe the calibration procedure more clearly in the revised version.

- *Although the authors have included Section 2.4 on learning $\beta(x^s)$ as a Markov random field, this section is not connected to other parts of the manuscript but only adds confusion. It is unclear what the authors meant by modeling $\beta(x^s)$ as a Marko random field. Does this mean that the authors no longer use Equation (6) to estimate $\beta(x^s)$? What are the assumptions of this Markov random field (MRF)? What are the parameters in this MRK and how is this MRK fitted?*

A3: The $\beta$ parameters are still computed with equations (6) and (7), but in addition to just computing a mean field approximation, we condition each vertex by the neighbors. This also imposes some smoothness on the posterior field of the $\beta$ parameters and and regularizes the problem. The fitting procedure is actually described on p. 8 l. 6-11 and in the caption of Fig. 2. Additionally, the conditioning on the neighbors is briefly explained in the text around p. 7 l. 27 - p. 8 l. 2. However, we agree that this description could be made clearer, and for this reason we will rewrite section 2.4 as needed. Regarding the parameters of the MRF, the MRF is over the $\beta$ parameters, and for the $\delta$ parameters we only obtain point estimates by fitting the parameters before and after obtaining the local $\beta$ values (amounting to a very short alternating optimization of $\beta$ and $\delta$). The smoothness of the fitting is controlled by the `dscale` parameter mentioned on p. 26 l. 12-15. The

MRF is fitted according to the procedure described in the caption of Fig. 2. We realize that even though how the fitting is exactly done is not so critical for how the *a posteriori* Gaussian process fields look like, this procedure should be more carefully explained, and not in a figure caption. We will integrate the description in the rewritten section 2.4.

- *It is also confusing how the parameters $\delta(x^s)$ are estimated.*

A4: The fitting of the $\delta$ parameters is carried out by optimizing them when computing the MRF as explained on p. 8 l. 12-18. While we think that the procedure is currently described in the text, it could be worded better, and we will do our best to also clarify this part of the text.

- *Line 14 of Page 8: " . . . finding $\hat{\beta}$ with Eq. (9) and (10), ..." Is this a typo? Should it be Eq. (6) and (7)?*

A5: Yes, this is a typo, which has now been fixed.

- *Page 8 Line 15: The objective function $\sum_{j=1}^{n}(m(x_\nu; \beta_\nu, \delta_\nu) - \psi_j)^2 + \sum_{j' \in \partial\nu}(\delta_\nu - \delta_{j'})^2$ and the optimization procedure are poorly explained. It should be noted that the mean function $m(\cdot; \cdot, \cdot)$ involved $\delta$ and $\beta$. It is very confusing how or why this function is used to estimate $\delta$ or $\beta$ individually or both of them jointly, and why it should be used this way.*

A6: This part of the text describes fitting the phase-shift parameters $\delta$, also mentioned above. For the "why" question, it is mentioned in the text that the nonlinear parameters cannot be calibrated the same way the $\beta$ parameters are dealt with. The first term blindly fits the mean function to data, while the second term imposes smoothness on the $\delta$-field. For simplicity and speed we don't use a dense error covariance matrix for the first term (as in ordinary least squares as opposed to generalized least squares), since for the $\delta$ parameters we are not interested in uncertainties. This is a modeling choice with which we aim to satisfy two objectives:

first, to get reasonable estimates of the $\delta$ field (for total column CO2 we expect that the spatial variation of the phase parameter should be be smooth) so that we do not end up fitting noise, and second, to perform this without computational complications. While computing $(m(x_\nu; \beta_\nu, \delta_\nu) - \psi_j)^T K^{-1}(m(x_\nu; \beta_\nu, \delta_\nu) - \psi_j)$ instead of the first term might be slightly more principled, this would come at the cost of needing to compute the Cholesky decomposition of the (often large) matrix $K$ at each optimization iteration. For large grids, the cost of finding the parameters would then be much higher. Regarding the loss in precision due to this compromise: with e.g. OCO-2 data, if there are observations that inform the calibration of the parameters at a certain location, there is a large number of those observations, and even if the full covariance matrix $K$ is used for the weighting, the parameters will end up being constrained enough that the most likely values from the generalized least squares would not differ by much from what is currently used. For this reason we are not concerned about the precision of our mean function.

The "how" part of the question was addressed in the comments above in A1, A3, and A4. We will still add a note about how the graph structure could be solved with algorithms such as generalized belief propagation, implementation of which is not yet included in satGP. This is future work that we hope to tackle in the near future.

As a final note we'd like to point out that the form of the mean function is generally data set specific, and it is the task of the modeler to understand the mean behaviour of the field before learning the GP parameters. While other data sets may require different, perhaps more complicated, mean function formulations, it is also possible to supply the mean function to satGP directly as an array.

2. *The notations in this manuscript are very confusing overall. For example, the authors sometimes use $\beta(^s)$ and later use $\beta_\nu$. The covariance parameters are even more confusing. There are $l, l_c,$ and $l_I$ . Even the definition of $I$ is not consistent: It is*

originally stated $I \subset \{x^s, x^t\}$, but later used as $I = ST$ or $I = S$, and $I = ST$. Also, the authors used $\Delta_{\text{year}}$ in Equation (11) and stated $\Delta_{\text{year}}$ is the duration of one year, does this mean $\Delta_{\text{year}} = 365$? Similarly, in Equation (15), the authors used $\Delta_{\text{period}}$; is it 365 as well?

A7: First, we agree that using both $\beta(x^s)$ and $\beta_\nu$ may be confusing. We use $\nu$ to refer to a generic vertex on a graph, whereas we use $\beta(x^s)$ on p. 7 l. 22 to underline that the $\beta$ parameters are space-dependent. We will remove this latter and explain the connection of the $\beta_\nu$ to the spatiality of the problem better.

Second, regarding the different $\ell$ variables, we'll do our best to make the notation more consistent. Due to the many length scales and dimensions in the problem, there is, however, a need to use some kind of subindex notation to differentiate between them. Despite this, the reviewer is correct to point out that more clarity is needed.

Third, regarding the index set notation with the letter $I$, we agree that this is not optimal, and that the notation is not consistent (there is e.g. both $ST$ and $I_{ST}$ etc.). We will do our best simplify the notation and improve the readability by minimizing redundancy and explaining what the terms stand for in all cases.

Fourth, the $\Delta_{\text{year}}$ vs. $\Delta_{\text{period}}$ is an intentional discrepancy: we use a period length of one year for the OCO-2 data (this is a modeling choice) but for other data products other period lengths could be used. For instance the OCO-3 is on the International Space Station and therefore its orbit is not Sun-synchronous and the local time varies: for this reason a period of one day could be used to model the diurnal variation. Since satGP is intended to be a general purpose software, we describe the covariance functions in generic terms and for this reason would like to keep the $\Delta_{\text{period}}$ notation. However, we will mention that the period can be changed and that $\Delta_{\text{year}}$ is a modeling choice for OCO-2 data.

3. *The authors suggest the multi-scale covariance function given in Equation (18):* $k(x, x'; \theta) = \delta(x, x')\sigma_x^2 + k_{\text{per}}(x, x'; \theta, I_S) + k_M(x, x'; \theta) + k_{\text{exp}}(x, x'; \theta, I + ST) +$

$k_W(x, x'; \theta, I)$.

- *First, I am not sure multi-scale is an accurate way to describe this covariance function. I feel this function is to add different types of covariance functions together, but these components not necessarily differ in terms of scales.*

A8: It is true that the combined covariance function works by adding different covariance functions together. However, how we decided to call the combined covariance function had a lot to do with the intended use of satGP: in the OCO-2 case we are particularly interested in finding the different length scales in the data induced by both spatial sparsity and underlying processes. Furthermore, remote sensing data often describes data from processes that involve different various characteristic length scales, as presented in e.g. figure 9. We could of course call the full kernel "multi-component", but we would like to emphasize that we are in particular interested in the different length scales. Note, that even if the kernel components are of different types, they still may describe processes at different length scales. A non-multiscale kernel would arise in a situation, where a kernel utilizes, say, an exponential and a periodic kernel component with the same length scale parameters. Such usage, while possible, would likely be slightly unusual. For this reason we'd like to keep the terminology that we currently have. We will, however, add a note that the kernel could also be called "multi-component", and briefly explain the reasoning behind the multi-scale name.

- *The authors did not explain clearly the component $kW(x, x0; \theta, I)$. Although Equation (16) states it is equal to $k \exp(x_W, x'_W; \theta^W, ST)$, the authors fail to explain $x_W$ or the quantifies in Equation (17) especially, $l$, $l^t$, $l_\parallel$, and $l_\perp$, and how these parameters are chosen/estimated.*

A9: We agree that this explanation is not adequate. The subindexes $W$ are spurious in $x_W$, and those will be dropped, e.g. in (16). Also, we will clarify how the rotated

kernels function and rephrase this part of the text to improve clarity. As with other covariance kernels, also these parameters may be found by maximum likelihood. This procedure is outlined in Section 2.7, but we will add a note that it applies also to the wind-informed kernel parameters.

- *What will happen if there are missing data in wind velocity?*

A10: In case of OCO-2 (and with many other products), the wind data is included with the data files. The satGP code also includes running a Gaussian process for the wind data (and the output can then be utilized with $k_W$). Wind data may also be read from an external file. We will add a note about these capabilities in the text.

- *Why isn't there an I involved in the Matérn component $k_M(\cdot, \cdot; \cdot)$?*

A11: Yes, there should of course be. This will also make the Matern description consistent with how the other kernels are described.

- *For the exponential component, the definition given in Equations (12) and (13) are not clear. At least there are two ways to define this component:*

$$k \exp(x, x_0; \theta, I_{ST}) = \tau^2 \exp\left(-\left|\frac{x - x'}{l_{ST}}\right|^\gamma\right)$$

*or*

$$k \exp(x, x_0; \theta, I_{ST}) = \tau^2 \exp\left(-\left|\frac{x^s - x^{s'}}{l_s}\right|^{\gamma_s}\right) \exp\left(-\left|\frac{x^t - x^{t'}}{l_t}\right|^{\gamma_t}\right)$$

*dependent on whether the spatial and temporal components share the scale or exponent parameters. I don't know what the authors have used, and there is no justification of their choice.*

A12: Each temporal dimension has its own scale length parameter. This is what the subindex $c$ in the sum and also in the term $\ell_c$ in (12) refers to. The sum is over

Interactive
comment

the dimensions in the set $I$, and while we think this is quite clearly presented, we will still try to clarify. This means that the second version listed above is what is being used, with the caveat that the exponents $\gamma$ are the same. If needed, this restriction can of course be quite easily lifted. For the OCO-2 experiments the exponent 2 was used.

- *The authors need to provide a better description of these components in the covariance function and explain why they are identifiable based on their formulations and definitions. Also, it is necessary to clarify whether some parameters are the same or vary across these components, such as $\tau^2$, $\gamma$, and $l$.*

A13: We will clarify that parameters such as $\tau$ are different for each kernel component. They can be found from the data, as was shown in the OCO-2 case. Of course the reviewer is correct that parameters of an arbitrary set of kernels would not necessarily be identifiable. However, what set of kernel components are chosen, is up to the modeler and depends on the data used. In the synthetic experiments we show that length scales of even three kernel components are recoverable, even though some parameters were slightly overestimated. We did perform additional tests, according to which parameters of two-component kernels are recoverable without such overestimation. We will add a comment on the modeler's role in picking the set of kernel components, underline that the synthetic studies verify the identifiability of the parameters, and furthermore do our best to improve the description of the kernels in general.

4. *I find Sections 2.6 and 2.7 quite difficult to understand. It seems that the authors use local kriging, that is, using a subset of data close to a prediction location $x^*$ to estimate the covariance parameters and to make prediction.*

A14: This is correct. We use a set of hyperspheres in the space of the inputs $x$, within which we fit the kernel parameters.

*Furthermore, it appears that the authors use different subsets of data to estimate the components in the covariance function. Why not using a single subset data to estimate the entire covariance function? Or, were the authors trying to avoid identifiability issue by using different data sets to estimate different covariance components? If a subset of data are used, I assume the size of this chosen subset is not too large, but why is there a need to use a block diagonal matrix $\tilde{K}$ as in Equation (22)? This approximation is not clearly explained, neither is $E_{ref}$ in Equation (22). 2 ?*

A15: We use the same subset of data to fit all the components at once, otherwise we could hardly claim that the parameters we choose are somehow optimal or correct. The sequentiality of the observation selection is due to something different: when we choose the (one and only) set of observations for fitting covariance parameters, we need to pick them so that all the (expected) length scales are represented in the data set. For instance, if the length scales are 10 kilometers and 1000 kilometers, we need to include both local dense data, and data from further away: if for instance we only include the closest observations, we don't really have leverage to say much about the longer-lengthscale behavior. We would like to point out more generally, that parameter identifiability is conditional on the data, so with some data (for instance with only one or zero observations) there will always be identifiability issues. While we think that we actually do explain what $E_{ref}$ is on p. 12 l. 24, we agree that the description is short, and that the block-diagonality is explained only implicitly (or not at all). We will clarify these points and include a better description of the $\tilde{K}$ matrices in the revised manuscript.

*Moreover, in Equation (19), should it be $> \sigma_{min}$ rather than $< \sigma_{min}$?*

A16: This is definitely true and has now been fixed.

5. *The authors mentioned the nearest neighbor Gaussian process, but did not cite the reference correspondingly.*

A17: Thank you for pointing this out, we will of course add a proper reference.

6. *It is unclear where or why MCMC is needed and how it is implemented (prior spec-ification etc.). The authors described optimization in Section 2 and also in the first paragraph of Page13. However, later in Page 17, the authors stated that MCMC is used instead. Section 2 does not describe MCMC.*

A18: The likelihood for learning the covariance parameters is noisy due to the observations selected changing with changing parameter values. For this reason optimization algorithms tend to get stuck in local minima. This is actually mentioned on p. 17 l. 4-5. We do mention that an Adaptive Metropolis implementation is included in the code, and that that can be used for finding the parameters (p. 13 l. 1-5). It is true that the priors are not described. We use flat priors, and will add information about them in the text in sections 4.1 and 4.4. We will also add a short description of MCMC to section 2.7.

7. *It should be Matérn covariance function, instead of Matern.*

A19: This has been fixed.

**Anonymous Referee #2**

A20: We thank Anonymous Referee #2 for appreciating our work. (No corrections or clarifications were requested.)

**Executive Editor Comment**

. . . *Therefore please provide the satGP v0.1 code or provide the reasons why the code can not be made publicly available in your revised submission to GMD.*

A21: We thank the executive editor for pointing out the code availability policy. We will make sure that the final revision conforms to the journal policies as requested.

---

## Author Response (AR1)

**gmd-2019-156: responses to reviewer comments**

We thank the anonymous reviewers and the editor for carefully reading the manuscript and for providing the very valuable comments. We address the comments one by one below. The reviewer comments are pasted verbatim below in italics, and the author responses to these comments can be found immediately under the comments, starting "A:". These are followed by "**Changes to manuscript:**" sections, where the line numbers refer to the diff file unless stated otherwise. Line numbers in the "A:" sections generally refer to the old version of the manuscript. Line numbers in the "Changes to manuscript" section refer to line numbers in the diff file.

**Anonymous Referee #1**

*This manuscript describes a model to analyze large spatio-temporal data. Although analyzing remote sensing data of enormous sizes is no double important and challenge, the manuscript fails to describe the model and its computation details and properties sufficiently or clearly. Please see below my comments that are not necessarily ordered chronologically or by importance:*

A: We thank the reviewer for this sincere assessment. To clarify the text and improve readability, we have restructured and rewritten large parts of the manuscript. This includes almost all of Sect. 2 (Methods), where the text has also been expanded in many places to more explicitly explain the technical details, with an emphasis on the requests made by Anonymous Referee #1. To aid the reader with the large number of different symbols in the manuscript (some of which were changed for clarity) we have added a full page list of symbols to help the reading. To illustrate the basic capabilities of satGP better, especially to those readers who are not so familiar with Gaussian process regression, we have added a short application to synthetic WACCM-generated ozone data, so that the reader can compare satGP output and uncertainties to the true underlying field, and appreciate that satGP is not a CO2-specific tool. We also fixed a few inaccuracies and minor bugs in the code, which lead to an increase in the version number of the software, from 0.1 to 0.1.2. Figures 1-2 and 8-10 were redone with this newest version of satGP.

*1. This manuscript suggest using the mean function of a particular form when analyzing OCO-2 data: $m(x; \beta, \delta) = f(x^t; \delta(x^s))\beta(x^s)$ This mean function is not a linear form of unknown parameters $\{\delta(x^s), \beta(x^s)\}$, noting that they are both dependent (i.e., varying) across locations. I find the description on how to estimate $\delta(x^s)$ and $\beta(x^s)$ extremely confusing.*

- *In Lines 10-20 of Page 6, it states that $\beta$ will be estimated using the formula of generalized least squared as given in Equation (6), and $\delta$ will be calibrated, but no explanation is given on how $\delta$ will be calibrated. In addition, the authors did not explain the dimension of the matrices $F$ and $K$ in Equation (6). Are they large so that $K^{-1}$ or $(F^T K^{-1} F)^{-1}$ difficult to compute?*

A1: First, in the earlier manuscript version we mention that we find a point estimate for the $\delta$ parameters before calibrating $\beta$ with generalized least squares, and that we then still one more time calibrate the $\delta$ parameters. We agree that the wording could be better, and we now clarify the alternating optimization in the sentence under (7) for the revised manuscript, adding that we use optimization algorithms for the task. We also give a reference to a later section for the full description of the procedure. Second, the reviewer is right about that the matrices are too large for direct inversion. For this reason the full size of the matrix $K$, and by extension the computing the dense matrices mentioned above would be prohibitively expensive. In our work the size of $K$ is up to order of $10^8 \times 10^8$, and such matrices would not fit to any computer's memory. **Changes to manuscript:** p. 7 l. 15, p. 8 l. 17-18, p.8 l. 19-24 (and the full section 2.3.2)

- *How is $\beta(x^s)$ estimated for a location $x^s$? For a location $x^s$,test without data/observation, can we estimate $\beta(x^s, test)$ and how?*

A2: The $\beta^s$ is estimated via the Markov Random Field, by fitting the parameters to match the mean function to local observations, and by conditioning on the parameter values at neighboring spatial locations. When there is no data nearby, the values of the parameters will be determined by prior values (if any – we use a flat prior) and the parameters at neighboring nodes in the MRF. We agree that the description in Section 2.4 is at the moment not very clear, and we will describe the calibration procedure more clearly in the revised version. **Changes to manuscript:** We have added section 2.3.2 detailing learning $\beta$ and $\delta$ for a given location, diff p. 13, last line – p. 15 l. 7.

- *Although the authors have included Section 2.4 on learning $\beta(x^s)$ as a Markov random field, this section is not connected to other parts of the manuscript but only adds confusion. It is unclear what the authors meant by modeling $\beta(x^s)$ as a Marko random field. Does this mean that the authors no longer use Equation (6) to estimate $\beta(x^s)$? What are the assumptions of this Markov random field (MRF)? What are the parameters in this MRK and how is this MRK fitted?*

A3: (Line numbers here refer to the old version of the manuscript) The $\beta$ parameters are still computed with equations (6) and (7), but in addition to just computing a mean field approximation, we condition each vertex by the neighbors. This also imposes some smoothness on the posterior field of the $\beta$ parameters and regularizes the problem. The fitting procedure was actually described on p. 8 l. 6-11 and in the caption of figure 2. Additionally, the conditioning on the neighbors was briefly explained in the text around p. 7 l. 27 - p. 8 l. 2. However, we agree that this description could be made clearer, and for this reason we have rewritten section 2.4 adding a lot of previously missing detail. Regarding the parameters of the MRF, the MRF is over the $\beta$ parameters, and for the $\delta$ parameters we only obtain point estimates by fitting the parameters before and after obtaining the local $\beta$ values (amounting to a very short alternating optimization of $\beta$ and $\delta$). The smoothness of the fitting is controlled by the `dscale` parameter mentioned on p. 26 l. 12-15, and of course also by the covariance kernel used, which affects the observation selection. The MRF is fitted according to the procedure described in the caption of figure 2. We realize that even though how the fitting is exactly done is not so critical for how the *a posteriori* Gaussian process fields look like, this procedure should be more carefully explained, and not in a figure caption. We will integrate the description in the rewritten section 2.4. (now 2.3) **Changes to manuscript:** The motivation behind the Markov Random Field paradigm is now explained in a separate subsection, 2.3.1, and learning the pointwise estimates, along with conditioning on neigbors, is now explained in the new section 2.3.2. The assumptions of the MRF are discussed first on p.12 l.9-10 and then on p.12 l.20-24. Spatial order of learning the graph is now explained on p.13 l.1-4, and elsewhere in that section.

- *It is also confusing how the parameters $\delta(x^s)$ are estimated.*

A4: The fitting of the $\delta$ parameters is carried out by optimizing them when computing the MRF as was explained on p. 8 l. 12-18. While we think that the procedure was described in the text, it could have been worded better, and we will do our best to also clarify this part of the text. **Changes to manuscript:** The $\delta$ parameter fitting has been included in the new section 2.3.2, particularly in the procedure p. 14 l.11 - p.15 l.3.

- *Line 14 of Page 8: " . . . finding $\hat{\beta}$ with Eq. (9) and (10), ..." Is this a typo? Should it be Eq. (6) and (7)?*

A5: Yes, this is a typo, this has been fixed.

- *Page 8 Line 15: The objective function $\sum_{j=1}^{n}(m(x_\nu; \beta_\nu, \delta_\nu) - \psi_j)^2 + \sum_{j' \in \partial\nu}(\delta_\nu - \delta_{j'})^2$ and the optimization procedure are poorly explained. It should be noted that the mean function $m(\cdot; \cdot, \cdot)$ involved $\delta$ and $\beta$. It is very confusing how or why this function is used to estimate $\delta$ or $\beta$ individually or both of them jointly, and why it should be used this way.*

A6: This part of the text describes fitting the phase-shift parameters $\delta$, also mentioned above. For the "why" question, it is mentioned in the text that the nonlinear parameters cannot be calibrated the

same way the $\beta$ parameters are dealt with. The first term blindly fits the mean function to data, while the second term imposes smoothness on the $\delta$-field. For simplicity and speed we don't use a dense error covariance matrix for the first term (as in ordinary least squares as opposed to generalized least squares), since for the $\delta$ parameters we are not interested in uncertainties. This is a modeling choice with which we aim to satisfy two objectives: first, to get reasonable estimates of the $\delta$ field (for total column CO2 we expect that the spatial variation of the phase parameter should be be smooth) so that we do not end up fitting noise, and second, to perform this without the need to handle covariances in the optimization. While taking to account observation covariances by computing e.g. $(m(x_\nu; \beta_\nu, \delta_\nu) - \psi_j)^T K^{-1} (m(x_\nu; \beta_\nu, \delta_\nu) - \psi_j)$ instead of plain squared error in the first term would be possible, we do not think that would really improve the fit for the $\delta$ parameters: this can be verified by e.g. looking at Fig. 1, which we have updated to show the fit to the actual observations instead of the daily means. Looking at that figure, it is clear that the phase shift $\delta$ parameters are correctly estimated. For this reason we are not concerned about the effect of this compromise to the precision of our mean function. Last, we'd like to emphasize that the covariances are properly accounted for when finding the $\beta$ parameters, so this compromize only affects the $\delta$ parameters.

The "how" part of the question was addressed in the comments above in A1, A3, and A4. We will still add a note about how the graph structure could be solved with algorithms such as generalized belief propagation, implementation of which is not yet included in satGP. This is future work that we hope to find time for at some point.

As a final note we'd like to point out that the form of the mean function is generally data set specific, and it is the task of the modeler to understand the mean behaviour of the field before learning the GP parameters. While other data sets may require different, perhaps more complicated, mean function formulations, it is also possible to supply the mean function to satGP directly as an array. **Changes to manuscript:** Optimization procedure is now explained much more carefully (p.14 l.11 - p.15 l.3). We also now mention the mean function in that section (p. 14 l. 5) to remind the reader of the context. Generalized belief propagation is mentioned as a possible future inference algorithm for the MRF on p.13 l.7

2. *The notations in this manuscript are very confusing overall. For example, the authors sometimes use $\beta(^s)$ and later use $\beta_\nu$. The covariance parameters are even more confusing. There are $l, l_c,$ and $l_I$ . Even the definition of $I$ is not consistent: It is originally stated $I \subset \{x^s, x^t\}$, but later used as $I = ST$ or $I = S$, and $I = ST$ . Also, the authors used $\Delta_{\text{year}}$ in Equation (11) and stated $\Delta_{\text{year}}$ is the duration of one year, does this mean $\Delta_{\text{year}} = 365$? Similarly, in Equation (15), the authors used $\Delta_{\text{period}}$ ; is it 365 as well?*

A7: First, we agree that using both $\beta(x^s)$ and $\beta_\nu$ may be confusing. We use $\nu$ to refer to a generic vertex on a graph, whereas we used $\beta(x^s)$ on p. 7 l. 22 to underline that the $\beta$ parameters are space-dependent. We have removed this latter notation and explain the connection of the $\beta_\nu$ to the spatiality of the problem better.

Second, regarding the different $\ell$ variables, we'll do our best to make the notation more consistent. The reviewer is correct to point out that more clarity is needed. We have made the notation more consistent and added these to a table of symbols.

Third, regarding the index set notation with the letter $I$, we agree that this is not optimal, and that the notation is not consistent (there is e.g. both $ST$ and $I_{ST}$ etc.). We have now made the notation consistent and no longer needlessly give the $I$ variables as arguments of the covariance functions. We also explain this notation in the table of symbols.

Fourth, the $\Delta_{\text{year}}$ vs. $\Delta_{\text{period}}$ was an intentional discrepancy: we use a period length of one year for the OCO-2 data (this is a modeling choice) but for instance the now-added WACCM example uses also 1.5 and 2 year periods. We have therefore removed the $\Delta_{\text{year}}$ notation altogether. **Changes to manuscript:** We added a full-page table explaining the most often used symbols and their dimensions, p.11. We explain the $I$-related symbols on p.15 l.18-19,21-22, and also in the table of symbols. The $\ell$-symbols are clarified, e.g. p.15 l.18,20,25-26. The notation $\Delta_{\text{year}}$ has been removed. We now use $\triangleq$ to emphasize that an equation is a definition.

3. *The authors suggest the multi-scale covariance function given in Equation (18): $k(x, x'; \theta) = \delta(x, x')\sigma_x^2 + k_{\text{per}}(x, x'; \theta, I_S) + k_M(x, x'; \theta) + k_{\text{exp}}(x, x'; \theta, I + ST) + k_W(x, x'; \theta, I).$*

- *First, I am not sure multi-scale is an accurate way to describe this covariance function. I feel this function is to add different types of covariance functions together, but these components not necessarily differ in terms of scales.*

A8: It is true that the combined covariance function works by adding different covariance functions together. However, how we decided to call the combined covariance function had a lot to do with the intended use of satGP: in the OCO-2 case we are in particular interested in finding the different length scales in the data induced by both spatial sparsity and underlying processes. Furthermore, remote sensing data often describes data from processes that involve different various characteristic length scales, as presented in e.g. figure 9. While we could of course call the full kernel "multi-component", we would rather like to emphasize that we are specifically interested in the different length scales. Note, that even if the kernel components are of different types, they still may describe processes at different length scales. A non-multiscale kernel would arise in a situation, where a kernel utilizes, say, an exponential and a periodic kernel component with the same length scale parameters. Such usage, while possible, would likely be slightly unusual. For this reason we'd like to keep the terminology that we currently have. We will, however, add a note that the kernel could also be called "multi-component", and briefly explain the reasoning behind the multi-scale name. **Changes to manuscript:** We mention that multi-component could be an alternative name, p.18 l.6-9.

- *The authors did not explain clearly the component $kW(x, x0; \theta, I)$. Although Equation (16) states it is equal to $k \exp(x_W, x'_W; \theta^W, ST)$, the authors fail to explain $x_W$ or the quantifies in Equation (17) especially, $l$, $l^t$, $l_\parallel$ , and $l_\perp$ , and how these parameters are chosen/estimated.*

A9: We agree that this explanation is not adequate. We now clarify how the rotated kernels function and rephrase this part of the text to improve clarity. As with other covariance kernels, also these parameters may be found by maximum likelihood. This procedure is outlined in Section 2.7, but we will add a note that it applies also to the wind-informed kernel parameters. **Changes to manuscript:** We have rewritten the section explaining the wind kernels, p.16 l.17 - p. 17 l.15, and we now explicitly give formulas for $x_w$ and $\ell_\parallel$ and $\ell_\perp$, and explicitly list $\ell_t$ and $\ell$ in the parameters of $k_w$. We explain that the $\rho$ parameters may be learned like the other parameters (p.20 l.17)

- *What will happen if there are missing data in wind velocity?*

A10: In case of OCO-2 (and with many other products), the wind data is included with the data files. The satGP code also includes running a Gaussian process for the wind data (and the output can then be utilized with $k_w$). Wind data may also be read from an external file. We will add a note about these capabilities in the text. **Changes to manuscript:** We now mention how wind data may be read in and that it is a required input for $k_w$, p.17 l.13-15.

- *Why isn't there an $I$ involved in the Matérn component $k_M(\cdot, \cdot; \cdot)$?*

A11: Yes, there should of course be. However, we decided instead to remove the $I$ arguments from all the kernels, since changing the dimensions over which the covariance functions work requires changing the code. **Changes to manuscript:** We have removed the $I$ arguments from kernels in equations 14-17, p.15-16.

- *For the exponential component, the definition given in Equations (12) and (13) are not clear. At least there are two ways to define this component:*

$$k \exp(x, x_0; \theta, I_{ST}) = \tau^2 \exp\left(-\left|\frac{x - x'}{l_{ST}}\right|^\gamma\right)$$

*or*

$$k \exp(x, x_0; \theta, I_{ST}) = \tau^2 \exp\left(-\left|\frac{x^s - x^{s'}}{l_s}\right|^{\gamma_s}\right) \exp\left(-\left|\frac{x^t - x^{t'}}{l_t}\right|^{\gamma_t}\right)$$

*dependent on whether the spatial and temporal components share the scale or exponent parameters. I don't know what the authors have used, and there is no justification of their choice.*

A12: Each dimension has its own scale length parameter. This is what the subindex $c$ in the sum and also in the term $\ell_c$ in (13) refers to. The sum is over the dimensions in the set $I$, and while we think this is quite clearly presented, we will still try to clarify. This means that the second version listed above is what is being used, with the caveat that the exponents $\gamma$ are the same. If needed, this restriction can of course be quite easily lifted by modifying the code. For the OCO-2 experiments the exponent 2 was used. **Changes to manuscript:** We try to explain the notation better, p. 15 l.16-27. We underline that the dimensions are independent and have separate length scale parameters, p.15 l.24-25. We have changed the notation to contain less subscripts, e.g. $\xi^\gamma_{\ell_I} \Rightarrow \xi^\gamma_I$ in equation 13.

- *The authors need to provide a better description of these components in the covariance function and explain why they are identifiable based on their formulations and definitions. Also, it is necessary to clarify whether some parameters are the same or vary across these components, such as $\tau^2$ , $\gamma$, and l.*

A13: We now clarify that parameters such as $\tau$ are different for each kernel component. They can be found from the data, as was shown in the OCO-2 case. Of course the reviewer is correct that parameters of an arbitrary set of kernels would not necessarily be identifiable. However, what set of kernel components are chosen, is up to the modeler and depends on the data used. In the synthetic experiments we show that length scales of even three kernel components are recoverable, even though some parameters were slightly overestimated. We did perform additional tests, according to which parameters of two-component kernels are recoverable without such overestimation. We will add a comment on the modeler's role in picking the set of kernel components, underline that the synthetic studies verify the identifiability of the parameters, and furthermore do our best to improve the description of the kernels in general. **Changes to manuscript:** We clarify that the parameters differ over the different kernel components, by subscripting the parameter vectors $\theta$, p. 15 l.28, p.16 l.1,8,22, that $\gamma$ is shared across dimensions, p.15 l. 25, and that $\ell$ parameters are different for each dimension (p.15 l.24-25). We state that the combined covariance parameter vector is now called $\theta$, (p.18 l.2). We have added a note about the modeler's role in modeling the data (p.18 l.3-4). We have added a simple one-kernel synthetic example, Sect. 4.2, which shows that the techniques used for learning mean function and covariance function parameters produce very good-looking fields, and that the uncertainties are what should be expected, implying that the method for finding the covariance parameters is able to converge to a well-performing parameter estimate. (p.26 last line - p.29 l.8).

4. *I find Sections 2.6 and 2.7 quite difficult to understand. It seems that the authors use local kriging, that is, using a subset of data close to a prediction location $x^*$ to estimate the covariance parameters and to make prediction.*

A14: This is correct. We use a set of hyperspheres in the space of the inputs $x$, within which we fit the kernel parameters. **Changes to manuscript:** We have significantly expanded and revised/rephrased/rewritten both of these sections to improve readability.

*Furthermore, it appears that the authors use different subsets of data to estimate the components in the covariance function. Why not using a single subset data to estimate the entire covariance function? Or, were the authors trying to avoid identifiability issue by using different data sets to estimate different covariance components? If a subset of data are used, I assume the size of this chosen subset is not too large, but why is there a need to use a block diagonal matrix $\tilde{K}$ as in Equation (22)? This approximation is not clearly explained, neither is $E_{ref}$ in Equation (22). 2 ?*

A15: We use the same subset of data to fit all the components at once, otherwise we could hardly claim that the parameters we choose are somehow optimal or correct. The sequentiality of the observation selection is due to something different: when we choose the (one and only) set of observations for fitting covariance parameters, we need to pick them so that all the (expected) length scales are represented in the data set. For instance, if the length scales are 10 kilometers and 1000 kilometers, we need to include both local dense data, and data from further away: if for instance we only include the closest observations, we don't really have leverage to say much about the behavior over longer length scales. We would like to point out more generally, that parameter identifiability is conditional on the data, so with some data (for instance with only one or zero observations) there will always be identifiability issues. While we think that we actually do explain what $E_{ref}$ is on p. 12 l. 24, we agree that the description is short, and that the block-diagonality is explained only implicitly (or not at all). We will clarify these points and include a better description of the $\tilde{K}$ matrices in the revised manuscript.

**Changes to manuscript:** We underline that we use a single data set (p. 18 l.22-23). We now clarify the block-diagonality and the relationship between $K$ and $\widetilde{K}$ in the text (p.20 l.24-30) We also disambiguated the notation in (new) Sect. 2.6 and added a short algorithm (figure 4) to describe the observation selection. We mention that the $E_{ref}$ is a set of random points from the domain (p. 20 l.12-13)

*Moreover, in Equation (19), should it be $> \sigma_{min}$ rather than $< \sigma_{min}$?*

A16: This is definitely true and has now been fixed.

5. *The authors mentioned the nearest neighbor Gaussian process, but did not cite the reference correspondingly.*

A17: Thank you for pointing this out, we have now added a proper reference. **Changes to manuscript:** Added reference, p.4 l.9

6. *It is unclear where or why MCMC is needed and how it is implemented (prior specification etc.). The authors described optimization in Section 2 and also in the first paragraph of Page13. However, later in Page 17, the authors stated that MCMC is used instead. Section 2 does not describe MCMC.*

A18: While learning the mean function parameters $\beta$ and $\delta$ utilizes optimization with the BFGS algorithm, covariance parameters are learned using MCMC. The likelihood for learning the covariance parameters is noisy due to the observations selected changing with changing parameter values. For this reason optimization algorithms tend to get stuck in local minima. This was actually mentioned on p. 17 l. 4-5 (old version of MS). We do mention that an Adaptive Metropolis implementation is included in the code, and that that can be used for finding the parameters (p. 13 l. 1-5). It is true that the priors are not described. We use flat priors, and will add information about them in the text in sections 4.1 and 4.4. We will also add a short description of MCMC to section 2.7. **Changes to manuscript:** MCMC and the motivation and its relation to optimization are now explained in more detail on (diff) p.21 l.4-17

7. *It should be Matérn covariance function, instead of Matern.*

A19: This has been fixed.

**Anonymous Referee #2**

A20: We thank Anonymous Referee #2 for appreciating our work. (No corrections or clarifications were requested.)

**Executive Editor Comment**

*. . . Therefore please provide the satGP v0.1 code or provide the reasons why the code can not be made publicly available in your revised submission to GMD.*

A21: We received the approval for open-sourcing satGP today from MIT, and will include the source code of the newest version as a supplement to the final manuscript version, after first adding the license headers and copyright notices.

[revised manuscript text omitted]

$$k_w(x,x';\theta,I_w) \triangleq k_{exp}(x_w, x'_w; \tau, ST)\{\ell_\|, \ell_\perp, \ell_t\}, 2). \tag{17}$$

The parameter $\rho$ in $\theta_w$ defines how strongly the magnitude of the wind vector at the test input, $w^* \triangleq [w^*_{lat}, w^*_{lon}]^T$ (the last parameter in $\theta_w$), affects the shape of the covariance. The kernel itself is an exponential kernel, where the spatial components of the vectors $x$ and $x'$ are transformed by wind data, and where the covariance lengths are transformed by wind speed. A spatio-temporal vector $x = [x^{lat}, x^{lon}, x^t]$ is transformed by wind to the vector $x_w$ in a new coordinate system according to

$$x_w \triangleq \begin{pmatrix} (x^s - x^{*s})^T w^\| \\ (x^s - x^{*s})^T w^\perp \\ x^t \end{pmatrix}, \tag{18}$$

where  $x^{\rm s}$ and $x^{*\rm s}$ are the spatial components of vectors $x$ and $x^*$, and where $w^{\parallel}$ and $w^{\perp}$ are the unit vectors in the lat-lon coordinates along and perpendicular to  wind direction at the test input $x^*$.

The spatial scaling  ($\ell$) parameters for $k_{\rm w}$, corresponding now to the covariance scales parallel and perpendicular to the wind direction, are given by

$$]1 + |w^*|\rho, \qquad \ell^{\perp} =_{\perp} \triangleq \ell,.$$

(19)

[revised manuscript text omitted]

---

## Author Response (AR2)

**GMD-2019-156: Responses to reviewer comments**

June 13, 2020

We thank the editor for the comments and for reading the manuscript one more time. We address the comments one by one below. The comments are pasted verbatim below in italics, and the author responses to these comments can be found immediately under the comments, starting "A:". These are followed by "**Changes to manuscript:**" sections, where the line numbers refer to the diff file unless stated otherwise.

**Editor comments**

1. P 5, L 10: *The expression "Observations generated by the Gaussian process" puzzles me. I thought the observations, used for training the GP, come from outside the GP, namely from measurements (as stated 2 lines before). Certainly it is possible to generate pseudo-observations once a GP is defined. But is this meant here?*

   A: This was due to confusion between real and synthetic data in the wording, even though one could think that also the non-synthetic fields had been generated by some (possibly very complicated) Gaussian process. Since this seems to cause confusion. We rephrase the sentence to improve clarity.

   **Changes to manuscript:** P 5 L 13-15 The sentence now reads: "Observations at locations $\{x_i^{\mathrm{obs}} : i = 1, \ldots, n\}$, both real and synthetic ones generated by the Gaussian process, are denoted by $\psi_i^{\mathrm{obs}} \in \mathbb{R}$, and the vector of all $\psi_i^{\mathrm{obs}}$ is written $\psi^{\mathrm{obs}}$."

2. P 5, L 20: *"contain" (plural).*

   A: Fixed

3. P 5, L 27/28: *I find this description misleading. Strictly speaking, the misleading starts right at eq. 1, where we have a function of x on the lhs, but a more complex function depending on x and another coordinate x' on the rhs. Unfortunately, to write k(x,x';...) seems to be general use in the GP literature. But when you now say "variables at x and x'", the misleading is deepened. I think, the relevant random variable is still psi(x), and at x.*
   *k(x,x';...) must be a scalar (representing the variance of psi at x), such that x is the relevant point, but x' is like a variable of integration (i.e. the sigma at x is something like integral[k(x,x';...)... dx']. However, a few lines later, when the infinite-dimensional GP is reduced to a n-dimensional problem, the writing k(x,x'...) makes sense and is understandable. Thus it seems possible to avoid misunderstanding by earlier mentioning of the transition to a finite problem.*

   A: We actually think, that while the notation $\Psi(x)$ in Eq. (1) is often used, it's misleading, and even in Table 1 we use $\Psi$ instead of $\Psi(x)$. We therefore remove the $(x)$ from (1). That should clarify the situation already quite a bit.

   About $k(x, x')$: The covariance function $k(x, x')$ gives the *co*variance between two points (but still a scalar, as is pointed out), and since $k(x, x') = k(x', x)$, I think $x$ and $x'$ are kind of equally important — even though notation-wise we usually are not interested in $x'$ when computing marginals (but then we usually also try to use $x^*$ instead of $x$). The (prior) *variance*, on the other hand, is actually given by $k(x, x)$. While we do mention the infinite-to-finite connection in the first sentence of Sect. 2.1 in passing, we agree that explaining the reduction to the finite-dimensional case further will help reading this section (and makes it intuitively clear what $k$ is).

**Changes to manuscript:** P 4 L 31 Removed $(x)$ from (1) LHS. P 5 L 2-4 Added sentence "The infinite-dimensional (since the domain of $x$ is typically infinite) description in Eq. (1) is below reduced to a finite-dimensional problem, in which case $k(x, x')$ describes an entry of the covariance matrix of the joint distribution of random variables $\Psi(x)$ over all $x$ that one is interested in." P 5 L 31 Added $\Psi(x)$ and $\Psi(x')$ to clarify the "... variables at $x$ and $x'$" sentence referred to in the comment. P 9 Added $\Psi(x)$ to the list of symbols.

4. P 6, L 18: *It is not clear what you mean. As delta and beta depend on the spatial coordinates, I would expect them to vary between inputs in $x^{obs}$, except when only the time coordinate varies in $x^{obs}$. And is it correct, that a solution must be found locally JUST BECAUSE $x^{obs}$ includes different locations? And finally, I do not see the connection to the next statement. Is local solution somehow equivalent to inversion of a full n x n matrix?*

A: Thank you for pointing out this error in the text. Parameters $\delta$ and $\beta$ of course do vary for each observation: while $\delta$ and $\beta$ are computed in a grid, values interpolated from that grid are used for constructing $F$. The reason for mentioning that a solution must be found locally was this variability, but that was actually not true and this sentence has been now removed. Solving local problems instead of the global one solves the problem that inverting the global huge $K$ is impossible. One of the anonymous referees requested in her/his comments that we comment on the sizes of the matrices, but still, it is true that the sentence was not connected well with the text around it. We changed the text and believe the sentence now fits in better.

**Changes to manuscript:** P 6 L 16-29 Erroneous statements have been removed. We now mention that each observation has different $\delta$ and $\beta$, and that the values are interpolated. We worked slightly on the text to include prior description in Eq. (6-7), since in satGP priors for $\beta$ can be used. We changed the wording on P 12 L 13,16 to reflect the possibility of using non-flat priors for $\beta$.

5. P 11, L 1: *What do you mean with "close in spatial covariance"?. Do you mean two points with quite high covariance (which can by geometrically far apart, perhaps due to teleconnection)?*

A: This means that we don't take to account parts of the covariance function dependent on time. We clarify the text.

**Changes to manuscript:** P 11 L 14 - P 12 L 2 The paragraph now reads: "Select $n_\nu$ observations $\psi_\nu^{\mathrm{obs}}$ of the observable that are close in terms of the spatial components of the covariance. Specifically, when evaluating whether to select an observation $\psi_i^{\mathrm{obs}}$ for carrying out computations at test input $x^*$ corresponding to some vertex $\nu$, we set the time component of $x_i^{\mathrm{obs}}$ to that of the test input, $x_i^{\mathrm{obs\,t}} \leftarrow x^{*\mathrm{t}}$, making the temporal part of the covariance function irrelevant in this selection process. Observation selection is described in detail in Sect. 2.5." .

6. P 12, L 20: *is gamma really a parameter CONTROLLING the exponent or is gamma simply the exponent? At least in eq. 13 it is an exponent. Please state also, what $x^c$ is (concrete examples are welcome). It is not listed in table 1.*

A: We clarify the sentence, add examples, and yes, $\gamma$ is the exponent.

**Changes to manuscript:** P 13 L 5-7 The sentence now reads: "... where $\gamma > 0$ is the exponent, $I$ is a (sub)set of the dimensions of the inputs $x$ and $x'$, $\ell_c$ are length scale parameters corresponding to the different dimensions in $I$, e.g. temporal ($\ell_{\mathrm{t}}$), zonal ($\ell_{\mathrm{lon}}$), or meridional ($\ell_{\mathrm{lat}}$) directions, and $x^c$ are different components of the inputs, e.g. $x^{\mathrm{t}}$, $x^{\mathrm{lat}}$, and $x^{\mathrm{lon}}$. ...".

7. P 24, L 12: *I wonder whether l_lat/lon have units, too, analogously to l_t. Are these degrees, radians, or something else? The same question goes to the entries in table 4.*

A: These are (unitless) distances on the unit ball, as was mentioned in Appendix A, so that would be radians on the equator. However, since "radians" could refer to different things in different parts of the globe, we prefer to talk about unit ball distances in general to avoid confusion. We have added this explanation also to the caption of Table 4.

**Changes to manuscript:** P 13 L 7-8 We added the sentence "The spatial length scale parameters $\ell_{\mathrm{lat}}$ are in units of distance on the surface of the unit sphere, corresponding to radians at the equator.". P 28 Add spatial length scale description in Table 4 caption.

8. P 28, L 6: *Is "simulation time" wall time here?*

   A: Yes, this is wall time.

   **Changes to manuscript:** P 28 L 19 "simulation time" ⇒ "simulation wall time"

**Other changes**

1. P 32 L 18-19 Code availability statement was modified to reflect that the code is now included as a supplement to the publication. (no coloring for this diff section)

[revised manuscript text omitted]

$$k_{\text{w}}(x, x'; \theta_{\text{w}}) \triangleq k_{\text{exp}}(x_{\text{w}}, x'_{\text{w}}; \tau, \{\ell_{\parallel}, \ell_{\perp}, \ell_{\text{t}}\}, 2). \tag{17}$$

The parameter $\rho$ in $\theta_{\text{w}}$ defines how strongly the magnitude of the wind vector at the test input, $w^* \triangleq [w^*_{\text{lat}}, w^*_{\text{lon}}]^T$ (the last parameter in $\theta_{\text{w}}$), affects the shape of the covariance. The kernel itself is an exponential kernel, where the spatial components of the vectors $x$ and $x'$ are transformed by wind data, and where the covariance lengths are transformed by wind speed. A spatio-temporal vector $x = [x^{\text{lat}}, x^{\text{lon}}, x^{\text{t}}]$ is transformed by wind to the vector $x_{\text{w}}$ in a new coordinate system according to

$$x_{\text{w}} \triangleq \begin{pmatrix} (x^{\text{s}} - x^{*\text{s}})^T w^{\parallel} \\ (x^{\text{s}} - x^{*\text{s}})^T w^{\perp} \\ x^{\text{t}} \end{pmatrix}, \tag{18}$$

where $x^{\text{s}}$ and $x^{*\text{s}}$ are the spatial components of vectors $x$ and $x^*$, and where $w^{\parallel}$ and $w^{\perp}$ are the unit vectors in the lat-lon coordinates along and perpendicular to wind direction at the test input $x^*$.

The spatial scaling ($\ell$) parameters for $k_{\text{w}}$, corresponding now to the covariance scales parallel and perpendicular to the wind direction, are given by

$$\ell_{\parallel} \triangleq \ell \sqrt[4]{1 + |w^*|\rho}, \qquad \ell_{\perp} \triangleq \ell. \tag{19}$$

The parameter vector for the exponential kernel then becomes $\theta_{\text{exp}} \leftarrow [\tau, \ell_{\parallel}, \ell_{\perp}, \ell_t, 2]^T$, where the last element denotes the exponent $\gamma$ used by the exponential kernel. A number of possible covariance ellipses resulting from the transformation procedure are shown in Fig. 3. Some data sets, like OCO-2, incorporate wind information, and satGP does have the capability of gridding that data using another Gaussian process. Reading in gridded wind data from other sources is also a possibility. Using $k_{\text{w}}$ requires that wind data at is available at each $x^*$.

The covariance functions used in this work to model $\Psi$ are sums of several kernels - sums of valid Gaussian process kernels remain valid kernels. The general form of the *multi-scale kernel* used in satGP is given by

$$k(x, x'; \theta) = \delta_{x,x'}\sigma_x^2 + \sum_{i=1}^{n_{\text{ker}}} k_{\text{ker}_i}(x, x'; \theta_{\text{ker}_i}), \tag{20}$$

where the first term, which in kriging is called the nugget, contains the observation error variances, and where each $\text{ker}_i \in \{\text{exp}, \text{M}, \text{per}, \text{w}\}$.

[Figure]

**Figure 3.** Equicovariance ellipses from the wind-informed kernel with various wind vectors $w^*$ and values of $\rho$. The wind values are taken at the test input $x^*$, but the covariance function $k$ is evaluated also for each pair of observations $x$ and $x'$.

The kernel components of a multi-scale kernel are in this work called *subkernels*. The combined set of parameters is denoted by $\theta = [\theta_{\ker_1}^T, \ldots, \theta_{\ker_{n_{\ker}}}^T]^T$. Not all subkernel types are included in all experiments – rather, the simulations in Sect. 4 utilize kernels with one to three components. What those components should be depends on what fields are being modeled and what kinds of correlation structures the user expects to find in the data. Section 4.1 discusses identifiability of the different subkernel
5   parameters of the multi-scale kernel.

Instead of calling $k(x, x'; \theta)$ in Eq. (20) a multi-scale kernel, the term multi-component kernel could also be used to describe the form. The term "multi-scale" underlines that the purpose of the combined kernel is to model well data, which contains several natural length scales, as remote sensing products often do. Furthermore, we believe that combining several kernels with identical length scale parameters does not represent a common use-case.

10  **2.5   Covariance localization and observation selection for the multi-scale kernel**

Using a large number of observations makes solving the Gaussian process Eq. (9) and (10) intractable as the cost of inverting the covariance matrix scales as $\mathcal{O}(n_{\mathrm{obs}}^3)$. This creates a need for finding approximate solutions while introducing as little error as possible. In satGP, covariance localization is used to utilize only a subset of observations for computing Eq. (9) and (10). To control the localization behavior the user needs to set two parameters: the maximum subkernel covariance matrix size $\kappa$ and
15  the minimum covariance parameter $\sigma_{\min}^2$.

Assume that the multi-scale kernel defined by the user contains $n_{\ker}$ subkernels. For each test input $x^*$ and for each subkernel $k_l$ the set of observations feasible for inclusion in $K$ in Eq. (6) and (7) is

$$A_{*,l}^{\mathrm{obs}} \triangleq \{\psi_i \in \psi^{\mathrm{obs}} | k_l(x_i^{\mathrm{obs}}, x^*) > \sigma_{\min}^2, \psi_i \notin A_{*,j}^{\mathrm{obs}} \, \forall j < l\}, \tag{21}$$

where the last condition prevents observations from being added by several subkernels. In the end we select a single set of
20  observations $A_*^{\mathrm{obs}}$ for each test input by combining some or all of the observations included in each $A_{*,l}^{\mathrm{obs}}$. The observation selection proceeds sequentially through the list of subkernels according to the procedure presented in Fig. 4. Recomputing the $\kappa'$ for each subkernel on line 3 of the algorithm allows selecting more than $\kappa$ observations by subkernels if the previous subkernels did not have $\kappa$ feasible observations available. This is done to allow the full kernel size to grow to $n_{\ker}\kappa$ when possible. On line 4, the observation selection operator $\mathcal{S}(A_{*,l}^{\mathrm{obs}}, \kappa')$ chooses $\kappa'$ observations from each $A_{*,l}^{\mathrm{obs}}$ either greedily by

**Data:** Set of feasible observations $A_{*,l}^{\text{obs}}$ for each
subkernel, maximum subkernel size $\kappa$,
observation selection operator $\mathcal{S}$.
**Result:** Set $A_*^{\text{obs}}$ of observations that are
informative for test input $x^*$

1  $A_*^{\text{obs}} \leftarrow \emptyset$ ;
2  **for** $i \leftarrow 1$ **to** $n_{\text{ker}}$ **do**
3     $\kappa' \leftarrow i\kappa - |A_*^{\text{obs}}|$ ;
4     $A_*^{\text{obs}} \leftarrow A_*^{\text{obs}} \cup \mathcal{S}(A_{*,i}^{\text{
[revised manuscript text omitted]